# *HOXBLINC* long non-coding RNA activation promotes leukemogenesis in NPM1-mutant acute myeloid leukemia

Ganqian Zhu[1,2,19], Huacheng Luo[3,19], Yang Feng[4], Olga A. Guryanova [4], Jianfeng Xu[5], Shi Chen[1,2], Qian Lai [3], Arati Sharma[6], Bing Xu[7], Zhigang Zhao[8], Ru Feng[9], Hongyu Ni[10], David Claxton[6,11], Ying Guo[2,12], Ruben A. Mesa[13,14], Yi Qiu [11,15], Feng-Chun Yang[2,12,14,16], Wei Li [17], Stephen D. Nimer[16,18], Suming Huang [3,11,20 ✉] & Mingjiang Xu [1,2,14,16,20 ✉]

*Nucleophosmin* (*NPM1*) is the most commonly mutated gene in acute myeloid leukemia (AML) resulting in aberrant cytoplasmic translocation of the encoded nucleolar protein (NPM1c$^+$). NPM1c$^+$ maintains a unique leukemic gene expression program, characterized by activation of *HOXA/B* clusters and *MEIS1* oncogene to facilitate leukemogenesis. However, the mechanisms by which NPM1c$^+$ controls such gene expression patterns to promote leukemogenesis remain largely unknown. Here, we show that the activation of *HOXBLINC*, a *HOXB* locus-associated long non-coding RNA (lncRNA), is a critical downstream mediator of NPM1c$^+$-associated leukemic transcription program and leukemogenesis. *HOXBLINC* loss attenuates NPM1c$^+$-driven leukemogenesis by rectifying the signature of NPM1c$^+$ leukemic transcription programs. Furthermore, overexpression of *HoxBlinc* (*HoxBlinc*Tg) in mice enhances HSC self-renewal and expands myelopoiesis, leading to the development of AML-like disease, reminiscent of the phenotypes seen in the *Npm1* mutant knock-in (*Npm1$^{c/+}$*) mice. *HoxBlinc*Tg and *Npm1$^{c/+}$* HSPCs share significantly overlapped transcriptome and chromatin structure. Mechanistically, *HoxBlinc* binds to the promoter regions of NPM1c$^+$ signature genes to control their activation in *HoxBlinc*Tg HSPCs, via MLL1 recruitment and promoter H3K4me3 modification. Our study reveals that *HOXBLINC* lncRNA activation plays an essential oncogenic role in NPM1c$^+$ leukemia. *HOXBLINC* and its partner MLL1 are potential therapeutic targets for *NPM1c$^+$* AML.

A full list of author affiliations appears at the end of the paper.

**N**ucleophosmin (*NPM1*) mutations are the most frequently recurring genetic abnormalities in patients with acute myeloid leukemia (AML), occurring in approximately 50% of adults and 20% of childhood AML with normal karyotypes[1,2]. *NPM1*-mutated AML has been included as a distinct AML entity in the World Health Organization (WHO) classification[3]. *NPM1* encodes a protein that is normally located in the nucleolus and has multiple functions such as biogenesis of ribosomes and maintenance of genomic stability[4]. *NPM1* mutations result in cytoplasmic mislocalization of the mutant protein ($NPM1c^+$), which is critical for its role in leukemogenesis[5,6]. As an AML-initiating lesion, $NPM1c^+$ maintains a distinctive transcriptional signature in AML cells, characterized by upregulation of *HOXA* and *HOXB* cluster genes and their oncogenic cofactor *MEIS1*[7,8]. However, the precise mechanisms by which $NPM1c^+$ drives the leukemic gene expression programs remain unclear.

Dysregulation of *HOXA/B* genes is a dominant mechanism of leukemic transformation and hematopoietic stem/progenitor cell (HSPC) deregulation[9]. A wide variety of molecular determinants including transcription factors, epigenetic regulators (e.g., polycomb and trithorax proteins), microRNAs, chromatin structure, and long non-coding RNAs (lncRNAs) are known to control *HOX* gene expression. However, their relationship with each other to fine-tune the *HOXA/B* gene expression pattern in AML remains to be elucidated. Recently two *HOXA/B* loci associated lncRNAs, *HOTTIP* and *HOXBLINC*, were shown to regulate transcription of *HOXA/B* genes through influencing epigenetic landscape[10–12]. *HOXBLINC* has been reported to play a critical role in hematopoietic specification during development through its *cis*-acting function to coordinate anterior *HOXB* gene expression via recruitment of the SETD1A/MLL1 histone H3K4 methyltransferase complexes[10].

Located in the anterior *HOXB* locus and serving as a regulator of *HOXB* gene transcription, the role of *HOXBLINC* in HSPC biology and leukemogenesis remains unknown. In this study, we demonstrate that *HOXBLINC* is upregulated restrictively in $NPM1c^+$ AML. *Npm1* mutant knock-in ($Npm1^{c/+}$) and *HoxBlinc*Tg HSPCs share significantly overlapping chromatin signatures and gene expression profiles in their upregulated genes as compared to WT HSPCs, including the $NPM1c^+$ signature *HoxA/B* cluster genes and homeobox oncogene *Meis1*. Transgenic overexpression of *HoxBlinc* lncRNA in hematopoietic cells led to the development of an AML-like disease by triggering HSC self-renewal and expanding myelopoiesis, similar to the phenotypes displayed by $Npm1^{c/+}$ mice, while inhibition of *HOXBLINC* in $NPM1c^+$ AML cells mitigates leukemogenesis. Importantly, *HoxBlinc* overexpression in HSPCs increases its binding to $NPM1c^+$ signature genes and drives the leukemic specific transcription program in HSPCs by recruiting the MLL1 complex to reorganize local chromatin signatures. Together, our studies provide compelling evidence for the potent oncogenic role of *HOXBLINC* in $NPM1c^+$-mediated leukemogenesis. *HOXBLINC* lncRNA and MLL1 could serve as potential therapeutic targets for the treatment of $NPM1c^+$ AML.

## Results

### *HOXBLINC* is specifically upregulated in $NPM1c^+$ AML. 
Upregulation of *HOX* genes, especially *HOXA* and *HOXB* cluster genes are not only a characteristic but also a dominant mechanism for the pathogenesis of AML[9]. *HoxBlinc* lncRNA has been shown to be required to activate anterior *HoxB* gene transcription during development[10]. To determine whether *HOXBLINC* is aberrantly expressed along with the *HOXB* genes in AML, we performed RT-qPCR on bone marrow mononuclear cells (BMMNCs) from a cohort of AML patients ($NPM1c^+$, $n = 25$; and *NPM1*-wt, $n = 40$;

see patient information in Supplementary Table 1) as compared to both BMMNCs ($n = 16$) and $CD34^+$ cells ($n = 11$) from healthy individuals. Interestingly, a dramatic upregulation of *HOXBLINC* was observed specifically in $NPM1c^+$ AML patients (Fig. 1a) as compared to *NPM1*-wt patients and normal $CD34^+$ cells. When the RNA-seq data from TCGA-LAML datasets consisting of a cohort of 181 AML patients was analyzed for *HOXBLINC* expression, significantly higher *HOXBLINC* expression was observed in $NPM1c^+$, but not *MLL*-rearranged ($MLLr^+$) AML patients as compared to $NPM1c^-MLLr^-$ patients (Fig. 1b). The expression of *HOXBLINC* was positively correlated with the expression levels of $NPM1c^+$ signature genes including anterior *HOXB* genes, *HOXA9* and *MEIS1*, but not *HOXB13* in this AML cohort (Supplementary Fig. 1a). Interestingly, AML patients with high *HOXBLINC* expression (the top thirty percentile of patients) had a significantly shortened survival as compared to patients with low *HOXBLINC* expression (the bottom thirty percentile, Supplementary Fig. 1b). Consistently, *HOXBLINC* was highly expressed in $NPM1c^+$ OCI-AML3 and IMS-M2 AML cells, but not *NPM1*-wt AML cells such as $MLLr^+$ MOLM-13, MV4-11, THP-1, NOMO-1, and OCI-AML2 cells, as well as $BCR-ABL1^+$ K562 and $JAK2V617F^+$ SET-2 cells (Supplementary Fig. 1c). These data collectively indicate that *HOXBLINC* is upregulated specifically in $NPM1c^+$ AML patients.

### Loss of *HOXBLINC* perturbs $NPM1c^+$-mediated transcription programs and leukemogenesis. 
To confirm *HOXBLINC* activation is a downstream event of $NPM1c^+$ and determine the role of *HOXBLINC* in $NPM1c^+$-mediated transcription regulation and leukemogenesis, we performed RNA-seq analysis on LSK cells isolated from the *Npm1* mutant knock-in ($Npm1^{c/+}$) mice[13]. As compared to WT LSK cells, $Npm1^{c/+}$ LSK cells had 871 downregulated genes and 980 upregulated genes, including *HoxBlinc* and the $NPM1c^+$ signature genes *HoxB2-5*, *HoxA7,9-11*, *Meis1*, and *Runx1* (Fig. 1c, Supplementary Fig. 1d), some of which are confirmed by qPCR (Supplementary Fig. 1e). Gene Ontology (GO) and gene set enrichment (GSEA) analyses revealed that the upregulated genes in $Npm1^{c/+}$ vs. WT LSK cells are enriched with cell fate commitment, cell cycle, myeloid cell proliferation, stem cell maintenance, Wnt and Jak-STAT signaling pathways, pathways in cancer, as well as AML *NPM1*-mutated and HOXA9 oncogenic pathway (Supplementary Fig. 1f, g).

We next examined the effect of *HOXBLINC* loss on $NPM1c^+$-mediated transcription regulation and leukemogenesis using $NPM1c^+$ OCI-AML3 cells. We created CRISPR-dCas9-KRAB mediated *HOXBLINC* epigenetic silencing clones (*HOXBLINC*i) by targeting the KRAB repressive domain to the *HOXBLINC* promoter (Supplementary Fig. 2a). We then compared genome-wide transcriptome changes between control and *HOXBLINC*i OCI-AML3 cells by performing RNA-seq analysis. Consistently, $NPM1c^+$ OCI-AML3 cells exhibited high expression of *HOXBLINC* lncRNA and the common $NPM1c^+$ AML signature genes (Fig. 1d). Interestingly, inhibition of *HOXBLINC* in OCI-AML3 cells significantly impaired the transcription of many $NPM1c^+$ signature genes such as *HOXB2-5*, *HOXA9-11*, *RUNX1*, and *MEIS1* (Fig. 1d). GSEA and GO analyses revealed that loss of *HOXBLINC* affects the pathways and genes involved in AML with *NPM1*-mutated, HOXA9 pathway, pathways in cancer, cell cycle, cell fate commitment, myeloid cell differentiation, and Wnt and JAK-STAT signaling pathways (Fig. 1e, f; Supplementary Fig. 2b), similar to those observed in the upregulated genes of $Npm1^{c/+}$ vs. WT LSK cells (Supplementary Fig. 1f, g).

We further explored whether *HoxBlinc* is required for $NPM1c^+$-mediated leukemogenesis by generating two *Tet-ON*

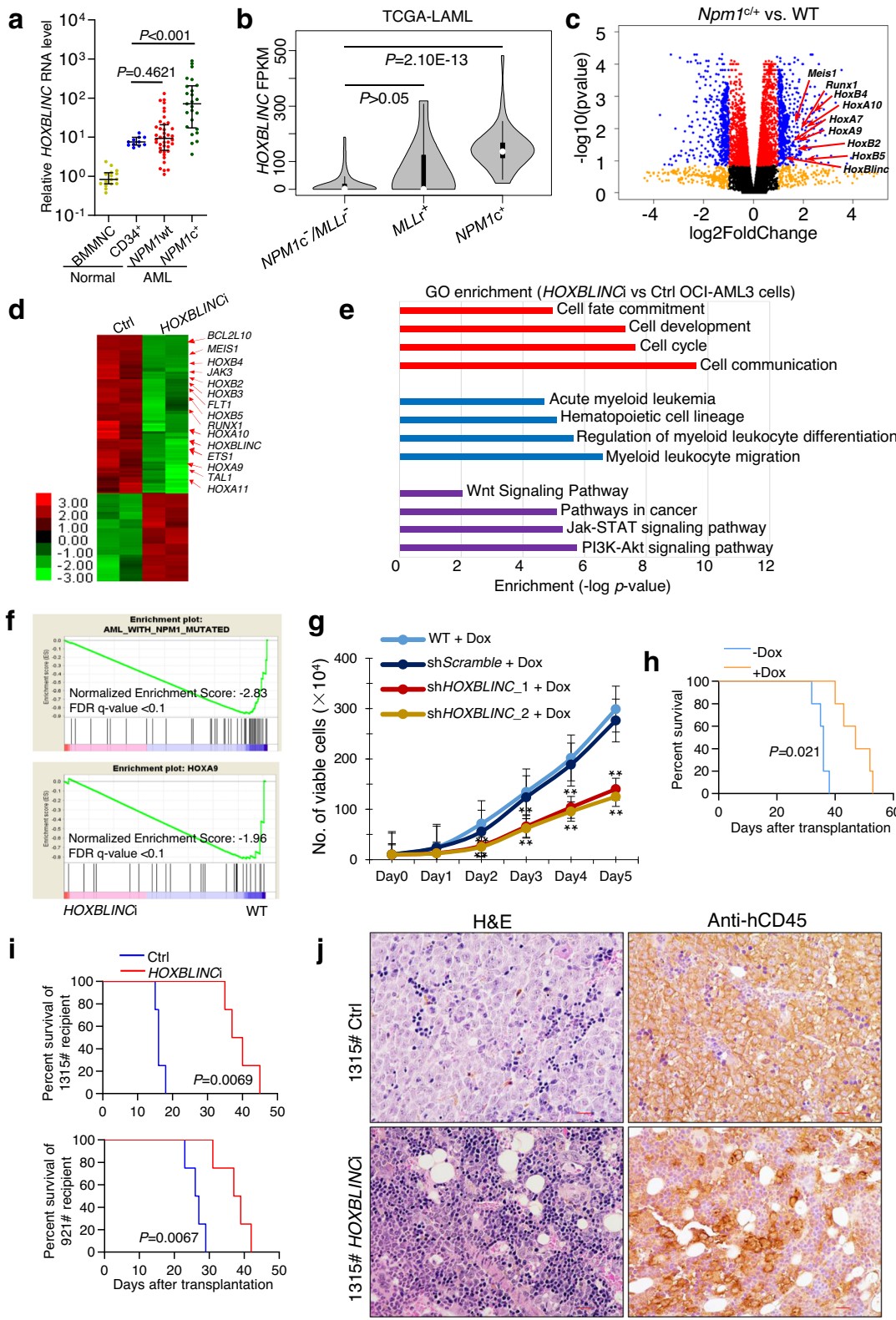

inducible *HOXBLINC* shRNA knockdown (KD) OCI-AML3 clones (Supplementary Fig. 2c). *HOXBLINC* KD significantly impaired OCI-AML3 cell proliferation compared to the scramble control (Fig. 1g). Cell cycle analysis revealed that *HOXBLINC* KD increased the sub-G0 cell population, suggesting *HOXBLINC* perturbation induced apoptosis (Supplementary Fig. 2d). When we transplanted a Dox-inducible *HOXBLINC* KD OCI-AML3 clone into *NOD-scid IL2Rγ[null]* (NSG) mice ($2 \times 10^5$ cells/mouse) followed by Dox induction or vehicle treatment, the Dox-treated recipient mice had significantly prolonged survival as compared to the recipients without Dox treatment (Fig. 1h). At 30 days after transplantation, Dox-treated mice had significantly lower hCD45[+] chimerism in

**Fig. 1 HOXBLINC is activated in NPM1c+ AML patients and loss of HOXBLINC perturbs NPM1c+-mediated transcription program and leukemogenesis.**
**a** RT-qPCR analysis of *HOXBLINC* RNA expression in BMMNCs ($n = 16$) and/or CD34+ cells ($n = 11$) from healthy individuals and in BMMNCs from *NPM1*wt ($n = 40$) or *NPM1c+* ($n = 25$) AML patients (Data is presented as Median ± interquartile; *P* value was calculated by two-sided Mann Whitney nonparametric test). **b** TCGA database (GSE62944) was used to retrieve RNA expression levels of *HOXBLINC* in BM cells from AML patients without *NPM1* mutations and *MLL* rearrangements (*NPM1c−/MLLr−*, $n = 128$), AML patients with *MLL* rearrangements (*MLLr+*, $n = 11$), or AML patients with *NPM1* mutations (*NPM1c+*, $n = 42$). Violin plots show mean, interquartile, and 1.5x interquartile. The width shows the probability density. *P* was calculated by two-tailed t-test. **c** Volcano plot of RNA-seq analysis of differentially expressed genes in *Npm1c+* knock-in (*Npm1c/+*) vs. WT LSK cells based on two independent experiments. **d** Heat map of RNA-seq analysis shows the up- and down-regulated genes in *HOXBLINC*-KRAB vs. WT OCI-AML3 cells based on two independent experiments. Arrows: dysregulated genes implicated in leukemogenesis. **e** The *HOXBLINC*-KRAB affected genes in OCI-AML3 cells were analyzed and annotated by the Gene Ontology (GO) analysis. **f** Enrichment of *HOXBLINC*-KRAB dysregulated genes in *NPM1*-mutated (*top*) and HOXA9 (*bottom*) oncogenic pathways by Gene Set Enrichment Analysis (GSEA). **g** Proliferation curves of WT, *shScramble*-expressing, and two different *shHOXBLINC*-expressing OCI-AML3 cells upon doxycycline (Dox) treatment for 5 days. (Bars represent the mean ± SD of total viable cells from three independent experiments; two-tailed t-test; **$P < 0.01$; $n = 3$ independent experiments). **h** Kaplan-Meier survival curve of inducible *shHOXBLINC* OCI-AML3 cell transplanted mice treated with or without Dox (Log-rank test; $n = 5$ mice/group). **i** Kaplan-Meier survival curves of mice transplanted with control or *HOXBLINC*-KRAB (*HOXBLINC*i) BM cells from 2 AML patients with *NPM1* mutation. 1315# patient (*NPM1c+;FLT3*wt, *left*), 921# patient (*NPM1c+;FLT3*mu, *right*). $n = 4$ mice/group. The Log-rank test was used to analyze differences between the survival curves. **j** Representative images of H&E (*right*) and anti-hCD45 immunostaining (*left*) of femur sections from 4 mice transplanted with control or *HOXBLINC*i BM cells from 1315# patients for 18 days (scale = 20 μm).

the BM, spleen and peripheral blood (PB) of the recipients compared to the untreated animals (Supplementary Fig. 2e). These results indicate that *HOXBLINC* KD suppresses OCI-AML3 leukemic cell proliferation both in vitro and in vivo, likely through the normalization of NPM1c+-induced abnormal gene expression patterns (Fig. 1d). In addition, we silenced *HOXBLINC* expression in primary AML cells with or without *NPM1c+* mutation (#1315: *NPM1c+; FLT3*wt, #921: *NPM1c+;FLT3mu*, #LPP4: *NPM1*wt;*MLLr+*) by the *CRISPR-dCas9-KRAB* (*HOXBLINC*i), and then xenografted them into NSG mice. Both #1315 and #921 exhibited high *HOXBLINC* expression, while #LPP4 had low *HOXBLINC* expression (Supplementary Table 1). In line with cell line xenograft results, mice receiving *HOXBLINC*i AML cells with *NPM1c+* mutations (#1315 and #921) had significantly prolonged survival as compared to mice transplanted with control cells (Fig. 1i). *HOXBLINC*i dramatically decreased the hCD45+ cell chimera in BM, spleen, and PB of these recipients (Supplementary Fig. 2f–h). Mice receiving control #1315 AML cells became moribund around 18 days after transplantation. Selected mice receiving either control or *HOXBLINC*i #1315 AML cells were sacrificed 18 days after transplantation to examine their human AML cell engraftment and AML development by combined FACS, immunohistochemical and morphological (cytospin) analyses on their BM cells. The engrafted human cells (hCD45+) in both control and *HOXBLINC*i recipients were positive for hCD33 and negative for CD19 and CD3, with a small fraction (~4%) being CD34+CD33low and over 50% being CD34low/+CD33+ (Supplementary Fig. 2f). Immunohistochemical analyses of BM sections also revealed robust infiltration by human CD45+ cells (Fig. 1j) and BM cytospins showed significant proportions of myeloid blasts (Supplementary Fig. 2g). These data demonstrated that the engraftment was from human AML cells but not multi-lineage human hematopoietic cells, and also clarified that moribund/death of these animals were caused by AML development and progression. In contrast, *HOXBLINC*i neither prolonged the survival nor decreased hCD45+ cell chimera in mice transplanted with *NPM1*wt;*MLLr+* (#LPP4) AML cells (Supplementary Fig. 2i). Thus, *HOXBLINC* perturbation decreased tumor burden and attenuated leukemic progression in vivo likely specific for *NPM1c+* AML patients.

**Transgenic expression of HoxBlinc in hematopoiesis leads to lethal AML-like disease in mice.** It has been shown that activation of a humanized *NPM1c+* knock-in allele in mouse HSCs (*NPM1c/+*) causes *Hox* gene overexpression, enhanced self-real, and expanded myelopoiesis, as well as the development of delayed-onset AML[14]. Since NPM1c+ activates *HOXBLINC* which is critical for

NPM1c+-mediated transcription program and leukemogenesis, it is important to determine whether *HOXBLINC* activation is sufficient to cause abnormal hematopoiesis and myeloid malignancies similar to the *NPM1c/+* mice. We first examined the *HoxBlinc* expression pattern along the HSC differentiation hierarchy. *HoxBlinc* expression was high in long-term (LT) and short-term (ST) HSCs, decreased in progenitor cells (MPP, CMP, and GMP), and was further decreased in the mature lineage cell populations except the B220+ B cells (Fig. 2a). The expression pattern of *HoxBlinc* in hematopoiesis suggests that this lncRNA might play an important role in regulating HSPC function. To investigate the impact of *HoxBlinc* activation on normal hematopoiesis and leukemogenesis in vivo, we generated a *HoxBlinc* transgenic (Tg) mouse model in which full-length mouse *HoxBlinc* cDNA was inserted into the mouse genome under the control of *Vav1* promoter and enhancer (HS321/45-*vav* vector) to ensure the expression of transgene specifically in hematopoiesis (Fig. 2b). Two founder *HoxBlinc*Tg mice were obtained. The transgene was inserted into the intron of the *Bin1* gene on chromosome 18q for Tg Line #1 (Fig. 2c). The expression levels of *HoxBlinc* RNA in BM cells were ~18- and 3-folds greater than the endogenous *HoxBlinc* expression in Tg Line #1 and #2, respectively (Fig. 2d).

Monitoring of a cohort of *HoxBlinc*Tg (Line #1) mice showed that within 1 year of age, 67% of *HoxBlinc*Tg mice (10 of 15) died or were killed because of moribund conditions, whereas none of the WT mice ($n = 12$) died (Fig. 2e). Moribund *HoxBlinc*Tg mice exhibited weight loss, hepatosplenomegaly, enlarged lymph nodes as well as pale footpads and femurs as compared to WT (Fig. 2f, Supplementary Fig. 3a). Peripheral blood examination revealed marked leukocytosis due to elevated immature myeloid cells and neutrophils, thrombocytopenia, and severe anemia in these moribund *HoxBlinc*Tg mice (Supplementary Fig. 3b). Morphologically, May–Grünwald–Giemsa stained PB smears showed significantly increased blasts (Fig. 2g, *left*). BM cell cytospin preparations also demonstrated a predominance of myeloid cells with increased immature myeloid precursors (Fig. 2g, *right*). Flow cytometric analyses of the BM cells revealed increased c-Kit+ (consistently > 20%) and Gr-1low immature myeloid cell populations, as well as decreased lymphoid and erythroid cell populations (Supplementary Fig. 3c, d). Morphologic evaluation of BM histologic sections revealed myeloid hyperplasia with increased immature myeloid precursors, which were myeloperoxidase (MPO) positive indicating myeloid origin (Fig. 2h, Supplementary Fig. 3e). In addition, histologic evaluation of the

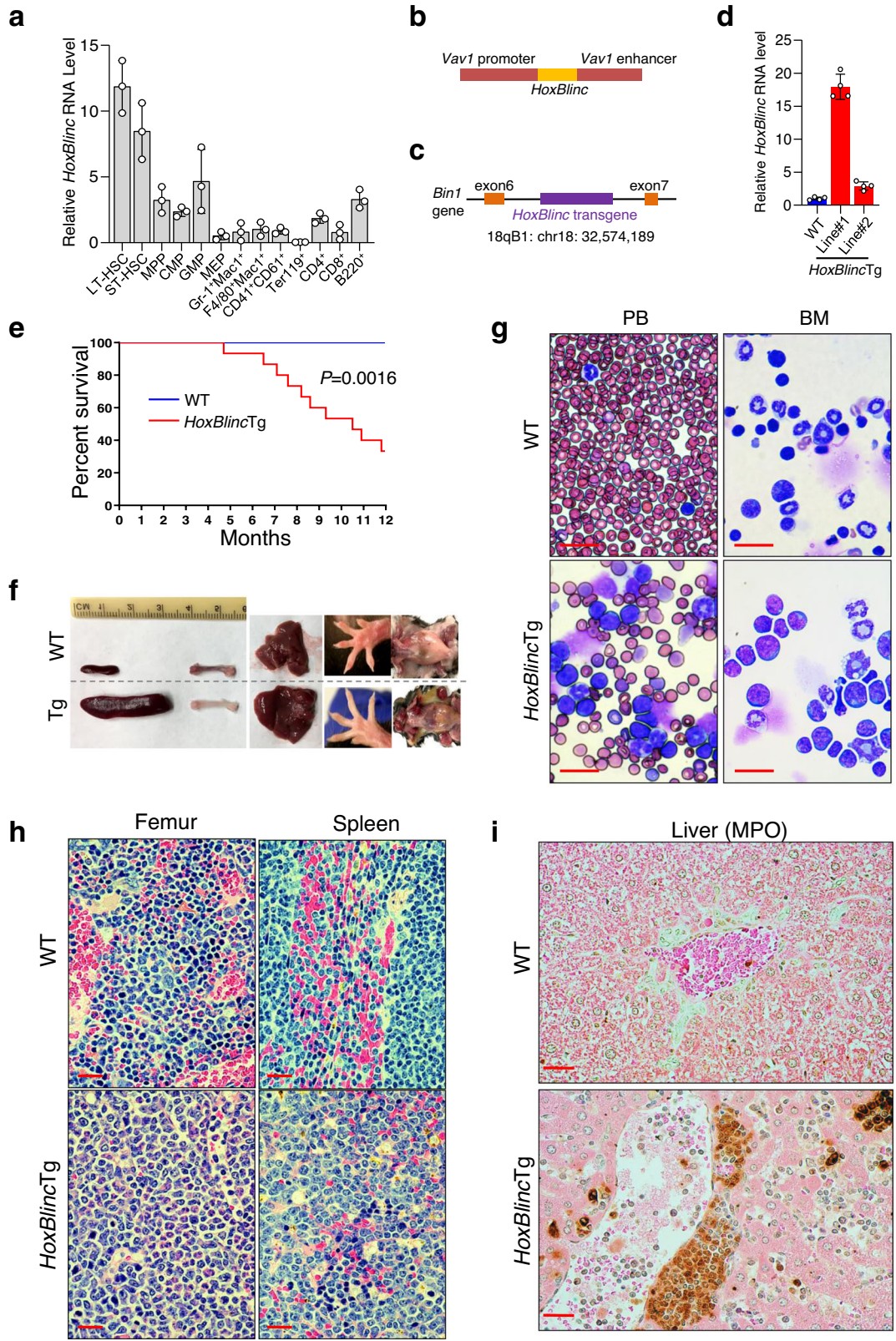

*HoxBlinc*Tg spleen, liver, and lymph node sections showed distortion of normal organ architecture with infiltration of MPO positive myeloid cells (Fig. 2h, i & Supplementary Fig. 3f). The sections of liver demonstrated a sinusoidal infiltration pattern with clusters of immature myeloid cells (Fig. 2i, Supplementary Fig. 3f). Consistently, Line #2 *HoxBlinc*Tg mice also developed

AML-like disease similar to Line #1, characterized by shortened survival and >20% c-Kit⁺ myeloid cells in the BM, PB, and spleen (Supplementary Fig. 3g–i). The longer survival exhibited in Line #2 than Line #1 *HoxBlinc*Tg mice suggests a dosage effect of the *HoxBlinc* expression on hematopoiesis and transformation. These data demonstrate that similar to *NPM1*ᶜ/⁺ mice, *HoxBlinc*

**Fig. 2 Transgenic overexpression of *HoxBlinc* in mice lead to lethal AML. a** RT-qPCR analyses of *HoxBlinc* expression levels on FACS-sorted BM cell populations along the hematopoiesis hierarchical differentiation tree of WT mice. Data are presented as the mean ± standard deviation (SD) of three independent experiments, each replicate utilized the flow-sorted populations of cells from combined BM cells of 7 WT mice. **b** Diagram of the *Vav1* promoter/enhancer driven *HoxBlinc* transgene strategy. **c** PCR-based transgenic integrating location identification (TAIL) assay maps the *Vav1-HoxBlinc* transgene integration site to the mouse chromosome 18 in Line #1 *HoxBlinc*Tg mice. **d** RT-qPCR analysis of the *HoxBlinc* RNA expression in BM cells of WT and 2 lines of *HoxBlinc*Tg mice. Data are presented as the mean ± SD of four independent experiments. **e** Kaplan-Meier survival curve of WT ($n = 15$) and *HoxBlinc*Tg (Line #1, $n = 15$) mice up to 1 year of age. *P* value was calculated by Log-rank test. **f** Gross appearance of spleens, femur, livers, feet, and lymph nodes of representative WT and moribund *HoxBlinc*Tg mice. **g** Representative images of May-Giemsa stained PB smears and BM cytospins prepared from 6 WT and 6 moribund *HoxBlinc*Tg mice. **h** Representative images of H&E stained femur and spleen sections of 6 WT and 6 moribund *HoxBlinc*Tg mice. **i** Representative images of MPO stained liver sections from 3 WT and 4 moribund *HoxBlinc*Tg mice. For panel (**g–i**), Scale = 20 μm.

overexpression in mice is sufficient to cause abnormal hematological characteristics resembling AML.

**Transgenic expression of *HoxBlinc* enhances HSC self-renewal and expands myelopoiesis.** The capacity of NPM1c[+] to promote enhanced HSC self-renewal and myeloid expansion has been well clarified[14]. To further understand the role of *HoxBlinc* in AML pathogenesis, we then determined the effect of *HoxBlinc* overexpression on HSPC function. Flow cytometric analyses on BM cells of young *HoxBlinc*Tg mice (8–10 weeks, Line #1) showed dramatically increased proportions of Gr1[+]/Mac1[+] granulocytic/ monocytic and B220[+] B cells as well as decreased proportions of CD4[+]/CD8[+] T and Ter119[+] erythroid cells as compared to age-matched WT mice (Supplementary Fig. 4a). Importantly, *HoxBlinc*Tg BM cells contained a significantly greater proportion of Lin[-]Scal-1[+]c-Kit[+] (LSK) cells, while Lin[−]Scal-1[−]c-Kit[+] (LK) cell population was similar as compared to WT mice (Fig. 3a). When the total number of LT-HSCs, ST-HSCs, and multipotent progenitor cells (MPPs) per femur were calculated based on their proportions within the LSK cell population and BM cellularity, the pools of both LT- and ST-HSCs, but not MPP were significantly expanded, although only the proportion of ST-HSCs within LSK cells were increased (Fig. 3b, Supplementary Fig. 4b). When each myeloid progenitor population was analyzed within the LK cells, a significantly higher percentage of GMP, but lower percentages of MEP/CMP cell populations were observed in *HoxBlinc*Tg mice than WT mice (Fig. 3c, Supplementary Fig. 4c). Consistent with the increased frequency of LSK and GMP, colony-forming unit cell (CFU-C) assays revealed significant higher frequencies of CFU-Cs, especially CFU-GM in the BM of *HoxBlinc*Tg mice than WT mice (Supplementary Fig. 4d). Similar to Line #1, significantly increased proportions of Gr1[+]/Mac1[+] and decreased proportions of erythroid cells were observed in the BM of Line #2 *HoxBlinc*Tg mice as compared to WT mice (Supplementary Fig. 4e). Line #2 *HoxBlinc*Tg BM cells also contained higher percentages of LSK and GMP cells than WT (Supplementary Fig. 4f, g). The similar hematological and disease phenotypes in both lines of *HoxBlinc*Tg mice indicate that these observed phenotypes are induced by the *HoxBlinc* transgenic expression but not the positional effect. Therefore, overexpression of *HoxBlinc* dysregulates HSPC pools with skewed hematopoiesis towards myeloid lineage in vivo.

We next examined the effect of *HoxBlinc* overexpression on the self-renewal and repopulation capacity of HSCs using in vitro replating and paired-daughter cell assays and in vivo competitive transplantation. A significantly higher replating potential was observed in each of the four successive rounds of replating in *HoxBlinc*Tg LSK cells than WT cells (Fig. 3d). Both symmetric and asymmetric cell divisions are required for the preservation of a normal HSC pool and continuous production of sufficient blood cells. Paired-daughter cell assays using WT and *HoxBlinc*Tg primitive CD34[−]LSK cells showed a higher proportion of *HoxBlinc*Tg CD34[−]LSK cells with symmetric self-renewal

capacity, while the cells that underwent symmetric differentiation or asymmetric self-renewal were reduced as compared to WT (Fig. 3e). Competitive transplantation assays showed that the donor cell (CD45.2[+]) chimerism in the PB of recipients transplanted with *HoxBlinc*Tg BM cells steadily increased, reaching ~80% 7 months after transplantation, while the CD45.2[+] cell population in the PB of mice receiving WT BM cells remained ~50% (Fig. 3f). Interestingly, mice receiving *HoxBlinc*Tg BM cells became moribund or died 2.5–7 months after transplantation (Fig. 3g). These *HoxBlinc*Tg BM recipients displayed similar hematological phenotypes as the primary *HoxBlinc*Tg mice, including high WBC counts, dramatically elevated immature myeloid cells and neutrophils, severe anemia, and decreased platelet counts (Supplementary Fig. 5a). Flow cytometric analyses of the donor (CD45.2[+]) and competitor (CD45.1[+]) derived BM cells revealed significantly greater percentages of c-Kit[+] and Gr-1[+]/Mac1[+] cells and lower proportions of CD3[+], B220[+] and Ter119[+] cells in *HoxBlinc*Tg vs. WT recipients (Supplementary Fig. 5b, c). Strikingly, the BM CD45.2[+] Lin[−] cells of *HoxBlinc*Tg recipients are comprised of significantly higher LK and LSK cells than that of CD45.1[+]Lin[−] cells in *HoxBlinc*Tg recipients and Lin[−] cells in WT recipients (Supplementary Fig. 5b). Consistently, a higher proportion of immature myeloid cells were observed in the BM of *HoxBlinc*Tg recipients (Supplementary Fig. 5d, e). The similar phenotypes displayed in recipients transplanted with *HoxBlinc*Tg BM cells and primary *HoxBlinc*Tg mice indicate that the aberrant HSPC function and AML-like disease induced by *HoxBlinc* overexpression in mice are transferable and HSPC cell-autonomous. Collectively, transgenic expression of *HoxBlinc* in mice enhances HSC self-renewal and expands myelopoiesis, leading to AML-like disease, reminiscent of the phenotypes seen in the *NPM1*[c/+] mice.

**Overexpression of *HoxBlinc* activates NPM1c[+] signature genes via increased enhancer/promoter chromatin accessibility in HSPCs.** To further delineate whether *HOXBLINC* is a downstream mediator of NPM1c[+] to maintain NPM1c[+]-signature gene activation in HSPCs, RNA-seq was performed with WT and *HoxBlinc*Tg LSK cells from 8-week old mice. Comparison of gene expression profiles of *HoxBlinc*Tg LSK cells to those of WT LSK cells identified 1,281 differentially expressed genes ($P < 0.05$ and fold-change ≥ 2.0, Fig. 4a). Among the DEGs, 718 were up-regulated and 563 were down-regulated. Strikingly, 27.3% of the up-regulated genes and 21% of the down-regulated genes were overlapped with the up- and down-regulated genes in *NPM1*[c/+] v.s. WT LSK cells, respectively (Fig. 4b). These common up-regulated genes of *HoxBlinc*Tg and *NPM1*[c/+] LSK cells (Supplementary Table 2) included the NPM1c[+] signature genes *HoxB2-5*, *HoxA9-10*, *Meis1*, and *Runx1*, which were confirmed by RT-qPCR (Supplementary Fig. 6a). Consistently, GO and GSEA analyses of the differentially expressed genes in *HoxBlinc*Tg vs. WT LSK cells revealed association/enrichment of similar transcription signatures and pathways with those in *NPM1*[c/+] vs. WT LSK cells, including *NPM1*-mutated

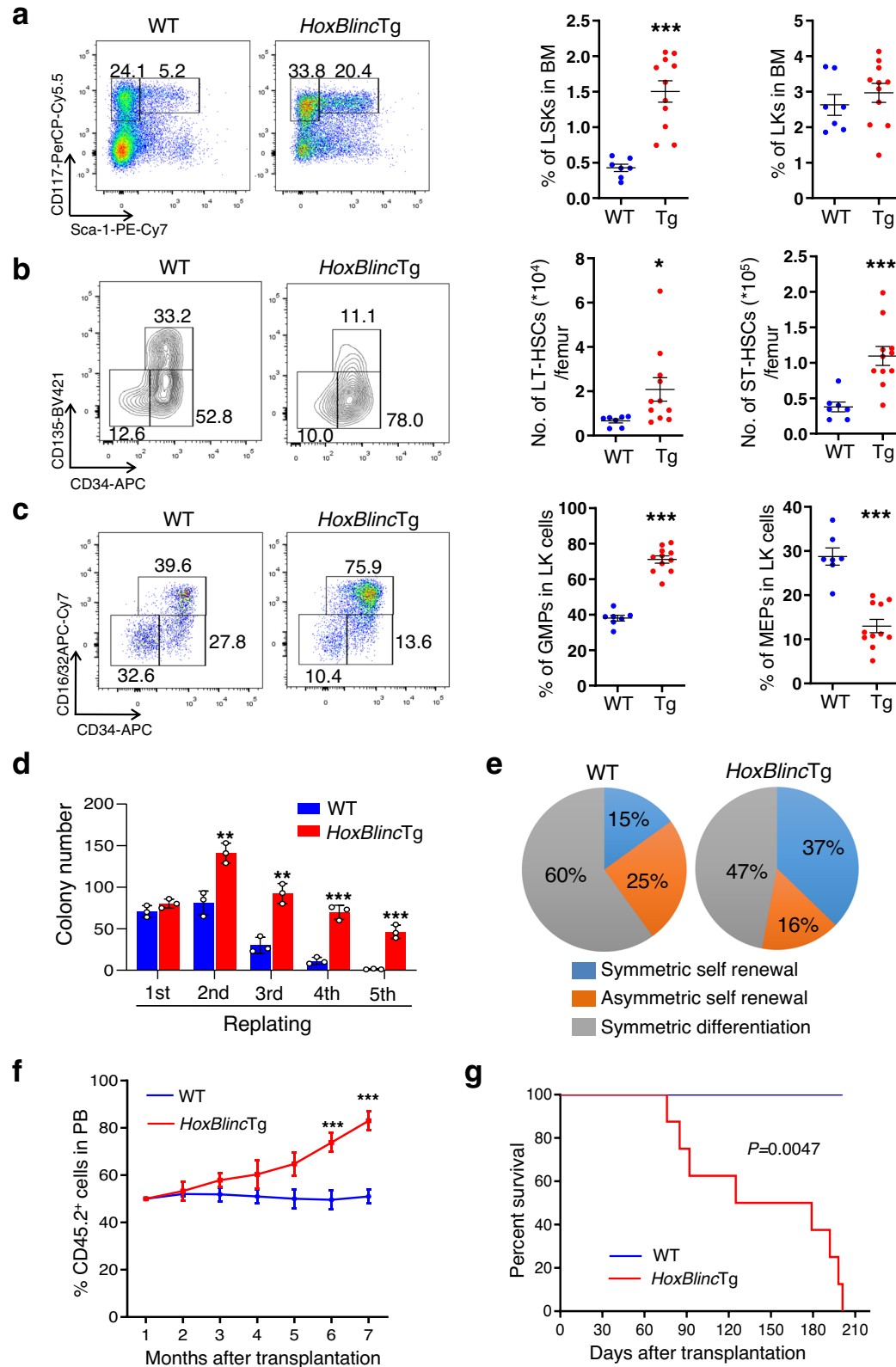

signature, HOXA9 oncogenic pathway, HSC proliferation, cell fate commitment, myeloid differentiation, Wnt and JAK-STAT signaling pathways (Fig. 4c, d, Supplementary Fig. 6b).

Since *HoxBlinc* promotes the expression of anterior *HoxB* genes by recruiting SETD1A and MLL1 complexes and

organizing active chromatin domain in the anterior *HoxB* locus[10], *HoxBlinc* upregulation caused by either NPM1c+ or *HoxBlinc*Tg could activate its target genes in HSPCs by accentuating enhancer/promoter chromatin accessibility. To test this, we carried out an assay for Transposase-Accessible

**Fig. 3 Transgenic expression of *HoxBlinc* enhances HSC self-renewal and expands myelopoiesis. a** FACS analysis of LSK (Lin⁻Sca-1⁺c-Kit⁺) and LK (Lin⁻Sca-1⁻c-Kit⁺) cell populations in the BM Lin⁻ cells of representative young (8–10 weeks old) WT and *HoxBlinc*Tg mice (*left*). Quantitation of the percent LSK and LK cells in the total BM cells of each genotype of mice is shown (*right*). **b** FACS analysis of LT-HSC, ST-HSC, and MPP cell populations in the BM LSK cells of representative young WT and *HoxBlinc*Tg mice (*left*). The total number of LT- and ST-HSCs per femur in WT and *HoxBlinc*Tg mice are shown (*right*). **c** FACS analysis of GMP, MEP, and CMP populations within BM LK cells of representative young WT and *HoxBlinc*Tg mice (*left*). Quantitation of the percent GMP and MEP cell populations in the BM LK cells of each genotype of mice is shown (*right*). (Data in (**a**–**c**) are presented as mean ± SEM; WT group $n = 7$; *HoxBlinc*Tg group n = 11; *$P < 0.05$ and ***$P < 0.001$ by two-tailed unpaired Student's t-test.) **d** The number of colonies per 100 WT or *HoxBlinc*Tg BM LSK cells are shown (1st). Colonies were replated every 7 days for 4 times (2nd-5th). (bars represent mean ± SD; $n = 3$ taken from three independent experiments; two-tailed t-test; **$P < 0.01$, ***$P < 0.001$). **e** Paired-daughter cell assays were performed on CD34⁻LSK cells clone-sorted from BM cells of WT and *HoxBlinc*Tg mice, and each cell was analyzed for symmetric self-renewal (blue), asymmetric cell division (orange), or symmetric differentiation (gray). **f** Kinetic flow cytometric analyses of CD45.2 chimerism in the PB of recipients transplanted with WT or *HoxBlinc*Tg BM cells. (bars represent mean ± SD; WT group $n = 5$; *HoxBlinc*Tg group $n = 8$; two-tailed t-test; ***$P < 0.001$). **g** Kaplan-Meier survival curve of recipient mice (8 mice/genotype) transplanted with WT or *HoxBlinc*Tg BM cells together with WT CD45.1 BMs (*P* value was calculated by Log-rank test).

Chromatin with high throughput sequencing (ATAC-seq) using LSK cells from WT, *NPM1^c/+^*, and *HoxBlinc*Tg mice (Supplementary Fig. 6c, d). Coincidently, *NPM1^c/+^* and *HoxBlinc*Tg LSK cells shared significant portions of genes exhibiting gain (28.9%) or loss (24.3%) of promoter accessibility (Fig. 4e, Supplementary Table 3). In addition, significant portions of the genes with promoter accessibility gain by either NPM1c⁺ or *HoxBlinc*Tg were also upregulated (Supplementary Fig. 6e). These genes included the NPM1c-signature genes *HoxB2-5, HoxA9-10, Meis1*, and *Runx1*, as well as other target genes such as *Stat1* and *Cdr2* (Supplementary Figs. 1d, 6f, 6g; Fig. 4f, g) that also play critical roles in HSC regulation and leukemogenesis. As a control, no significant changes in chromatin accessibility was observed in the *Lypla1* locus, and the expression of *Lypla1* was not altered by *HoxBlinc* upregulation caused by either NPM1c⁺ or *HoxBlinc*Tg (Supplementary Fig. 6g). These results suggest that overexpression of *HoxBlinc* lncRNA specifically activates NPM1c⁺ signature genes via enhancing enhancer/promoter chromatin accessibility in HSPCs.

***HoxBlinc* directly binds to target genes and mediates chromatin interactions to drive gene regulatory networks in HSPCs.** CTCF boundaries facilitate enhancer/promoter interactions within confined topologically associated domains (TADs). We recently reported that a CTCF boundary in the posterior HOXA locus establishes and maintains an active TAD to drive posterior *HOXA* gene expression[15]. To examine whether *HoxBlinc* overexpression affects CTCF defined anterior TAD domain and enhancer/promoter regulatory networks in the anterior *HoxB* locus, circular chromosome conformation capture using high throughput sequencing (4C-seq) was performed using the several *HoxB* locus CTCF binding sites (CBSs) as viewpoints in *HoxBlinc*Tg vs. WT Lin⁻c-Kit⁺ cells (Fig. 5a). When the CBS located between *HoxB4* and *B5* (CBS4/5, which overlaps *HoxBlinc* gene) was used as a viewpoint, CBS4/5 interacted with the +43Kb CBS (+43CBS, Fig. 5b). CBS4/5 also contacted each of the anterior *HoxB* genes (Fig. 5b), suggesting that either CBS4/5 or more likely *HoxBlinc* communicates with anterior *HoxB* gene promoters. When +43CBS was used as a viewpoint, the interaction of CBS4/5 and +43CBS was confirmed and +43CBS communicated with anterior *HoxB* genes too (Fig. 5b). Interestingly, *HoxBlinc* overexpression intensified each of these long-range interactions within the anterior *HoxB* locus mediated by CBS4/5 and/or +43CBS (Fig. 5b). In contrast, +73Kb CBS (+73CBS) and *HoxB13* CBS (CBS13) did not interact with the anterior *HoxB* genes, although +73CBS interacted with CBS5/6 and CBS8/9, which however was not affected by *HoxBlinc* overexpression (Fig. 5b). Furthermore, *HoxBlinc* overexpression also induced the long-range interactions of CBS4/5 and/or +43CBS with the promoter regions of *HoxBlinc* target genes such as *Stat1, Crd2*,

and posterior *HoxA* genes, but not non-*HoxBlinc* target *HoxD* genes (Fig. 5c, Supplementary Fig. 7a). These data indicate that *HoxBlinc* coordinated with the CBS4/5 and +43CBS in HSPCs to facilitate and maintain long-range chromatin interactions with NPM1c⁺ signature gene loci for their activation.

To completely understand the mechanism by which *HoxBlinc* overexpression regulates hematopoietic transcription program in HSPCs, we carried out ChIRP-seq (Chromatin Isolation by RNA Purification combined with deep sequencing) to map the genomic *HoxBlinc* binding sites in WT and *HoxBlinc*Tg Lin⁻c-Kit⁺ cells. Overexpression of *HoxBlinc* significantly increased its binding to the promoter regions of anterior *HoxB* genes and other *trans* targets, such as *Stat1, Cdr2, Wnt5a, Runx1,* and posterior *HoxA* genes (Fig. 5d, Supplementary Fig. 7b). The *HoxBlinc* binding to anterior *HoxB* and *Runx1* promoters were confirmed by ChIRP-qPCR (Supplementary Fig. 7c, d). The global *HoxBlinc*-binding site distribution in Lin⁻c-Kit⁺ cell genome revealed that *HoxBlinc* mainly interacted with noncoding regions, including intergenic regions, introns, and promoters. Emphatically, *HoxBlinc* overexpression markedly increased its occupancy with promoters and UTRs (Fig. 5e). Furthermore, GO analysis of *HoxBlinc*-bound genes found enrichment of pathways important for HSPC regulation and leukemogenesis such as *Hox* genes, AML, HSC proliferation, Wnt signaling, and cell cycle (Supplementary Fig. 7e). Integration of ChIRP-seq, RNA-seq, and ATAC-seq datasets from WT and *HoxBlinc*Tg HSPCs revealed that around 74% of the genes with increased *HoxBlinc* binding exhibited a ≥ 2 folds increase in gene expression levels (Fig. 5f) and 44.7% of them showed increased promoter chromatin accessibility (Fig. 5g). These data revealed that *HoxBlinc* acts as an epigenetic regulator to control target gene expression through remodeling promoter chromatin accessibility. Transcription motif analysis showed that the top *HoxBlinc* bound motifs in *HoxBlinc*Tg HSPCs are transcription factors important for hematopoiesis, such as CTCF and PU.1 (Supplementary Fig. 7f, Supplementary Table 4). These data demonstrated that *HoxBlinc* directly binds to hematopoietic specific target genes, mainly the NPM1c⁺ signature genes, and mediates long-range chromatin interactions to drive gene regulatory networks in HSPCs.

**Recruitment of MLL1 is critical for *HoxBlinc* overexpression mediated target gene expression and leukemogenesis.** As *HoxBlinc* recruits SETD1A and MLL1 complexes to organize active chromatin domain in the *HoxB* loci in mouse ECSs derived primitive erythroid progenitor cells[10], we performed RIP-qPCR analysis which showed much greater *HOXBLINC* enrichment in the immunoprecipitates of anti-MLL1 and anti-SETD1A, but not control IgG and anti-LSD1 antibodies in high *HoxBlinc* expressing OCI-AML3 cells as compared to the low *HoxBlinc* expressing OCI-AML2 cells (Supplementary Fig. 8a), confirming the interaction of human *HOXBLINC* with MLL1 and SETD1A. We next

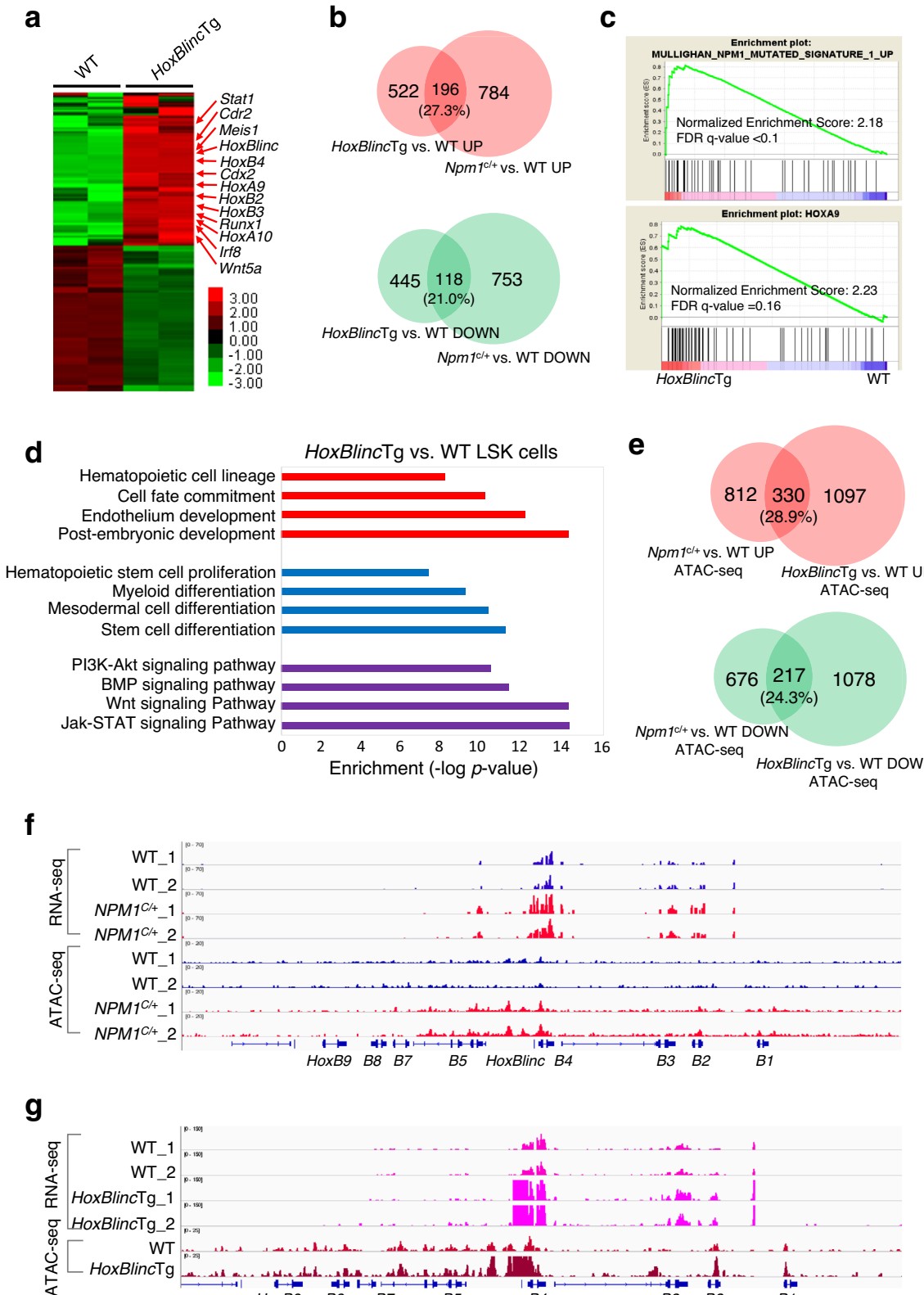

**Fig. 4 Overexpression of *HoxBlinc* activates *NPM1c*+ signature genes via enhancing promoter chromatin accessibility in LSK cells. a** Heatmap of RNA-seq analysis shows the up- and down-regulated genes in *HoxBlinc*Tg vs. WT LSK cells based on two independent experiments. Red arrows: up-regulated genes implicated in HSPC regulation/leukemogenesis. **b** Overlap of up- (*top*) or down- (*bottom*) regulated genes between *HoxBlinc*Tg vs. WT and *NPM1*c/+ vs. WT LSK cells. **c** Enrichment of upregulated genes involved in *NPM1*-mutated signature (*top*) and HOXA9 (*Bottom*) oncogenic pathway upon overexpression of *HoxBlinc* in LSK cells by GSEA. **d** The *HoxBlinc* overexpression dysregulated genes in LSK cells were analyzed and annotated by GO analysis. **e** Overlap of global gain (*top*) or loss (*bottom*) of promoter chromatin accessibilities between ATAC-seq data of *HoxBlinc*Tg vs. WT and *NPM1*c/+ vs. WT LSK cells. **f** RNA-seq (chromatin accessibility, *top 4 tracks*) and ATAC-seq (gene expression, *bottom 4 tracks*) analysis of WT and *Npm1*c/+ LSK cells in the *HoxB* gene locus. **g** RNA-seq analysis (*top 4 tracks*) and ATAC-seq analysis (*bottom 2 tracks*) of WT and *HoxBlinc*Tg LSK cells in the *HoxB* gene locus.

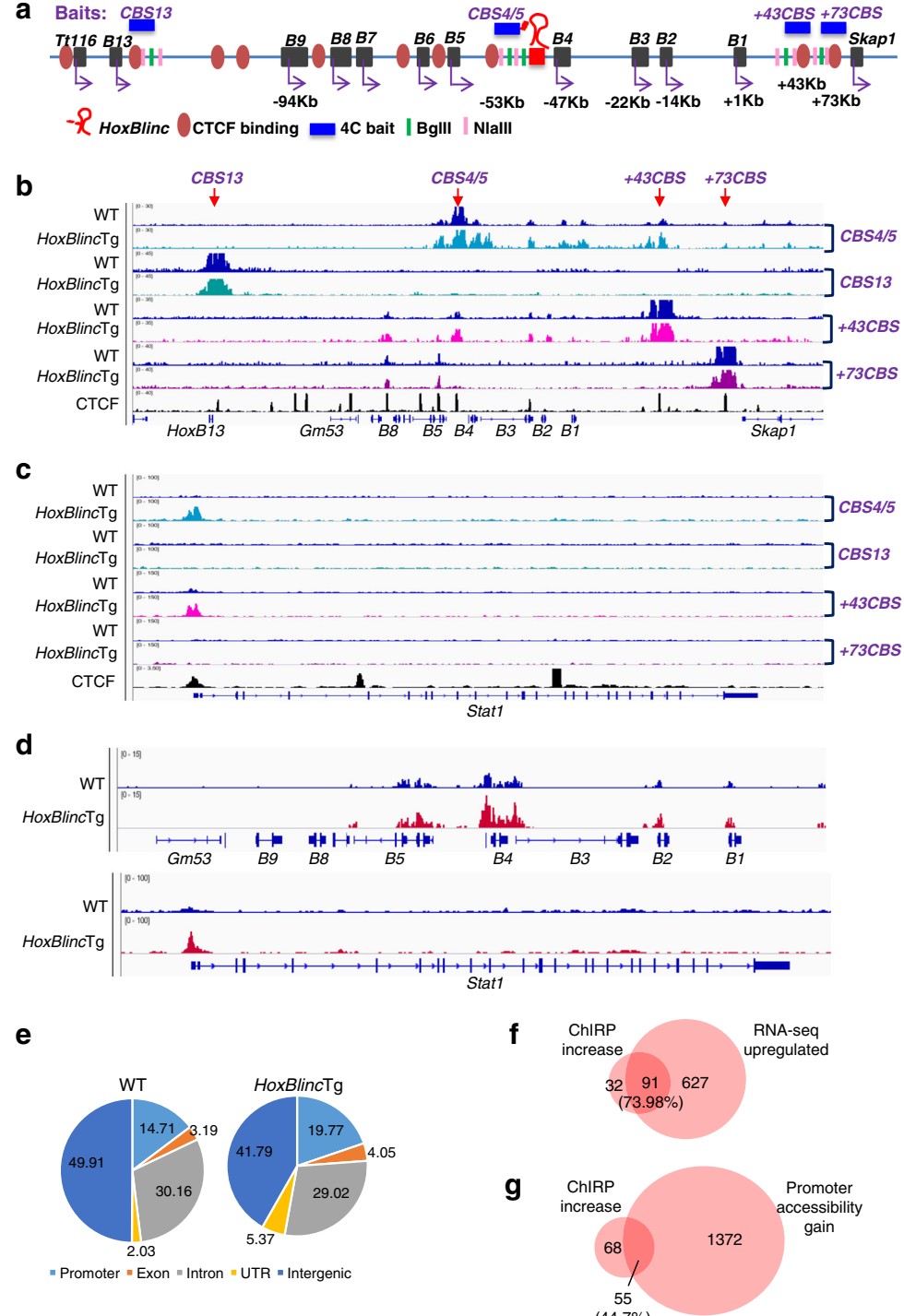

**Fig. 5 *HoxBlinc* directly binds to target genes and mediates chromatin interactions to drive gene regulatory networks in HSPCs. a** Schematic diagram showing the CTCF binding sites (CBS), location of the 4C baits in *HoxB* locus of the mouse genome. **b, c** Long-range chromatin interactions with *HoxB* locus (4 baits) as determined by 4C-seq analysis in WT and *HoxBlinc*Tg Lin⁻c-Kit⁺ cells. *HoxBlinc* overexpression increased the interactions of CBS4/5 and +43CBS with the *HoxB* genes (**b**), *HoxBlinc* overexpression also enhanced the interaction of CBS4/5 or +43CBS with *Stat1* promoter region (**c**), ChIP-seq analysis of CTCF binding sites was obtained from the NCBI GEO public database (GSM918748). **d** ChIRP-seq analysis of *HoxBlinc* bindings to the *HoxB* and *Stat1* gene loci in WT and *HoxBlinc*Tg Lin⁻c-Kit⁺ cells. **e** The pie chart shows the distribution of promoter, exon, intron, UTR, and intergenic region within the total *HoxBlinc* binding sites in WT (*left*) and *HoxBlinc*Tg (*right*) Lin⁻c-Kit⁺ cells as identified by ChIRP-seq. **f** Overlap of genes with *HoxBlinc* binding gain to their promoter regions as identified by ChIRP-seq and the upregulated genes as identified by RNA-seq in *HoxBlinc*Tg vs. WT HSPCs. **g** Overlap of genes with *HoxBlinc* binding gain to their promoter regions as identified by ChIRP-seq and genes with promoter accessibility gain as identified by ATAC-seq in *HoxBlinc*Tg vs. WT HSPCs.

examined whether SETD1A and MLL1 are important for *HoxBlinc* overexpression mediated abnormal HSPC function and leukemogenesis both in vitro and in vivo. However, knockdown *Mll1*, but not *Setd1a* in *HoxBlinc*Tg LSK cells was capable of mitigating the abnormal replating potential mediated by *HoxBlinc* overexpression (Fig. 6a, Supplementary Fig. 8b). Furthermore, the genetic deletion of one *Setd1a* allele in *HoxBlinc*Tg mice (*Setd1a*$^{+/-}$;*HoxBlinc*Tg) did not rescue the abnormal hematologic phenotypes induced by *HoxBlinc* overexpression in vivo (Supplementary Fig. 8c–f), indicating that SETD1A is unlikely to be a key player in *HoxBlinc* overexpression mediated abnormal hematopoiesis. When transplantation was performed using *HoxBlinc*Tg Lin⁻c-Kit⁺ cells transduced with lentivirus expressing scramble control or sh*Mll1*, *Mll1* KD significantly prolonged the survival of recipients receiving sh*Mll1*-expressing *HoxBlinc*Tg Lin⁻c-Kit⁺ cells as compared to sh*Scramble* expressing *HoxBlinc*Tg Lin⁻c-Kit⁺ cells (Fig. 6b). And the aberrant expansion of CD117⁺/CD11b⁺ immature myeloid cells and GMPs, as well as anemia in HoxBlincTg mice, were also largely restored by *Mll1* KD (Fig. 6c, d). These data indicate that *Mll1* KD is capable of mitigating the AML development induced by *HoxBlinc* overexpression.

Since MLL1 is critical for *HoxBlinc* overexpression mediated abnormal hematopoiesis, *HoxBlinc* overexpression in HSPCs might activate its target genes by increasing MLL1 recruitment and thereby enhancing H3K4me3 occupancy to facilitate enhancer/promoter chromatin accessibility. To confirm this, we carried out MLL1 and H3K4me3 ChIP-seq using WT and *HoxBlinc*Tg LSK cells. Combined ChIP-seq and CHIRP-seq analyses revealed significantly high overlap for the genomic binding sites of *HoxBlinc* and MLL1 (Fig. 6e–g, Supplementary Fig. 9a). Impressively, 51.2% of the genes with increased *HoxBlinc* binding exhibited elevated MLL1 recruitment and most of which (31.7%) also showed increased H3K4me3 occupancy, including the NPM1c⁺-signature genes such as anterior *HoxB*, posterior *HoxA*, *Meis1*, and *Runx1*, as well as other hematopoietic genes such as *Stat1* and *Cdr2* (Fig. 6e–g, Supplementary Fig. 9a, Supplementary Table 5). In contrast, no significant changes in MLL1 recruitment and H3K4me3 occupancy were observed in the control *Lypla1* (no change in *HoxBlinc* binding), *Eno1* (no *HoxBlinc* binding), and *Car6* (no *HoxBlinc*/MLL1 binding) loci (Supplementary Fig. 9b); and the expression of such genes was not altered by *HoxBlinc* overexpression. These data demonstrate that *HoxBlinc* overexpression mediates target gene expression via increased recruitment of MLL1 and subsequent enhancement of H3K4me3 occupancy.

## Discussion

In a humanized NPM1c⁺ knock-in mouse model, NPM1c⁺ enhanced HSC self-renewal and expanded myelopoiesis leading to 1/3ʳᵈ of the animals developing late-onset AML[14]. Brunetti et al. recently reported that specific reduction of NPM1c⁺ lessens key features of the leukemic program[16]. Thus, *NPM1* mutation is an AML-driving lesion and maintains leukemia mainly through a gain-of-function by the NPM1c⁺. NPM1c⁺-mediated leukemogenesis has been shown to depend on the unique gene expression signatures such as *HOXA/B* and *MEIS1* activation[17]. However, how this aberrant gene expression program is driven and maintained is largely unknown. In this study, we show that NPM1c⁺ deregulates its signature genes, perturbs hematopoiesis, and promotes leukemogenesis via the activation of a critical lncRNA, *HOXBLINC*. *HOXBLINC* overexpression is strongly associated with *NPM1* mutations in AML and NPM1c⁺ expression leads to *HoxBlinc* activation in HSPCs. Although *HoxBlinc* is involved in normal hematopoietic development[13], *HoxBlinc* overexpression plays an

essential and sufficient oncogenic role in NPM1c⁺-mediated signature gene expression, HSPC deregulation, and leukemogenesis. Therefore, our studies identify *HoxBlinc* activation as a critical downstream mediator for NPM1c⁺.

In addition to mutations and/or aberrant expression in protein-coding genes, misregulation of lncRNAs perturbs cellular physiology in multiple ways and plays important roles in the development and progression of various cancers[18]. However, how lncRNAs affect the initiation and progression of malignant myelopoiesis remains to be determined. Accordingly, lncRNAs have been profiled in various myeloid leukemias in order to identify potential oncogenic lncRNAs[19–21]. Recent studies have shown that *LncHSC-2* and *Hottip* lncRNAs contribute to the control of critical signaling pathways in HSC regulation[12,22]. However, a direct link between lncRNAs and oncogenesis remains elusive in malignant myelopoiesis. Furthermore, the detailed molecular mechanisms underlying lncRNA dysregulation-mediated myeloid malignancy development remain largely unknown. Our RNA-seq analyses on *HOXBLINC*i vs. control NPM1c⁺ OCI-AML3 cells and *HoxBlinc*Tg vs. NPM1$^{c/+}$ or WT LSK cells demonstrate that *HOXBLINC* regulates NPM1c⁺ signature genes including anterior *HOXB* genes where *HOXBLINC* resides and genes located on other chromosomes such as *HOXA9-10*, *MEIS1*, and *RUNX1*. ChIRP-seq analyses reveal that *HoxBlinc* binds to the promoters of both resident anterior *HoxB* cluster genes and distant target genes such as posterior *HoxA* and *Runx1*. These results indicate that the regulation of the NPM1c⁺ signature genes by *HoxBlinc* is achieved through the direct *HoxBlinc* binding via *cis* and *trans* actions. Indeed, GSEA and GO analyses signify enrichment of genes for NPM1-mutated signature, HOXA9 oncogenic pathway, Wnt and JAK-STAT signaling pathways in *HOXBLINC*i vs. control OCI-AML3 cells and *HoxBlinc*Tg or NPM1$^{c/+}$ vs. WT LSK cells. Additional functional analyses demonstrate that normal hematopoiesis requires a tightly controlled *HOXBLINC* expression, and its misregulation caused by NPM1c⁺ is an oncogenic event in leukemogenesis. Our findings on the direct oncogenic role of *HOXBLINC* in *NPM1*-mutated AML could serve as a blueprint for implicating lncRNAs in AML leukemogenesis.

LncRNAs show a high versatility in their mechanism-of-action, influencing many cellular processes such as spatial conformation of chromosomes, chromatin modifications, and RNA transcription[23]. *HoxBlinc* recruits MLL1 and SETD1a to anterior *HoxB* loci and controls their gene expression by regulating chromatin states during development[10]. Recently, SETD1A was demonstrated to act as a positive epigenetic regulator of HSC function during hematopoiesis[24,25]. However, genetic deletion of one *Setd1a* allele could not rescue *HoxBlinc* overexpression mediated abnormal HSPC function and leukemogenesis. On the other hand, our *Mll1* KD experiments revealed that loss of MLL1 mitigates *HoxBlinc* overexpression induced abnormal HSPC function and leukemogenesis in vivo, demonstrating MLL1 as a dependency in NPM1c⁺/ *HoxBlinc*-overexpressing AML. We further showed that *HoxBlinc* overexpression increases *HoxBlinc* occupancy (in *cis* and *trans* actions) and MLL1 recruitment at the promoters of its target genes including the NPM1c⁺ signature genes to induce aberrant long-range chromatin interaction networks and promote their expression. These results are in line with a recent study by Kuhn et al. showing that the chromatin binding of MLL1 is critical for *NPM1*-mutated leukemias and Menin-Mll1 interaction controls the expression of *HOX*, *MEIS1*, and *FLT3* genes in *NPM1*-mutated AML[17]. Interestingly, *NPM1* mutations are mutually exclusive with several genetic abnormalities such as the *MLL*-rearrangement and *MLL1*-partial tandem duplication in AML[26,27]. Thus, in *NPM1*-mutated AML, NPM1c⁺ achieves and maintains its signature gene expression program via *HoxBlinc* overexpression, which increases

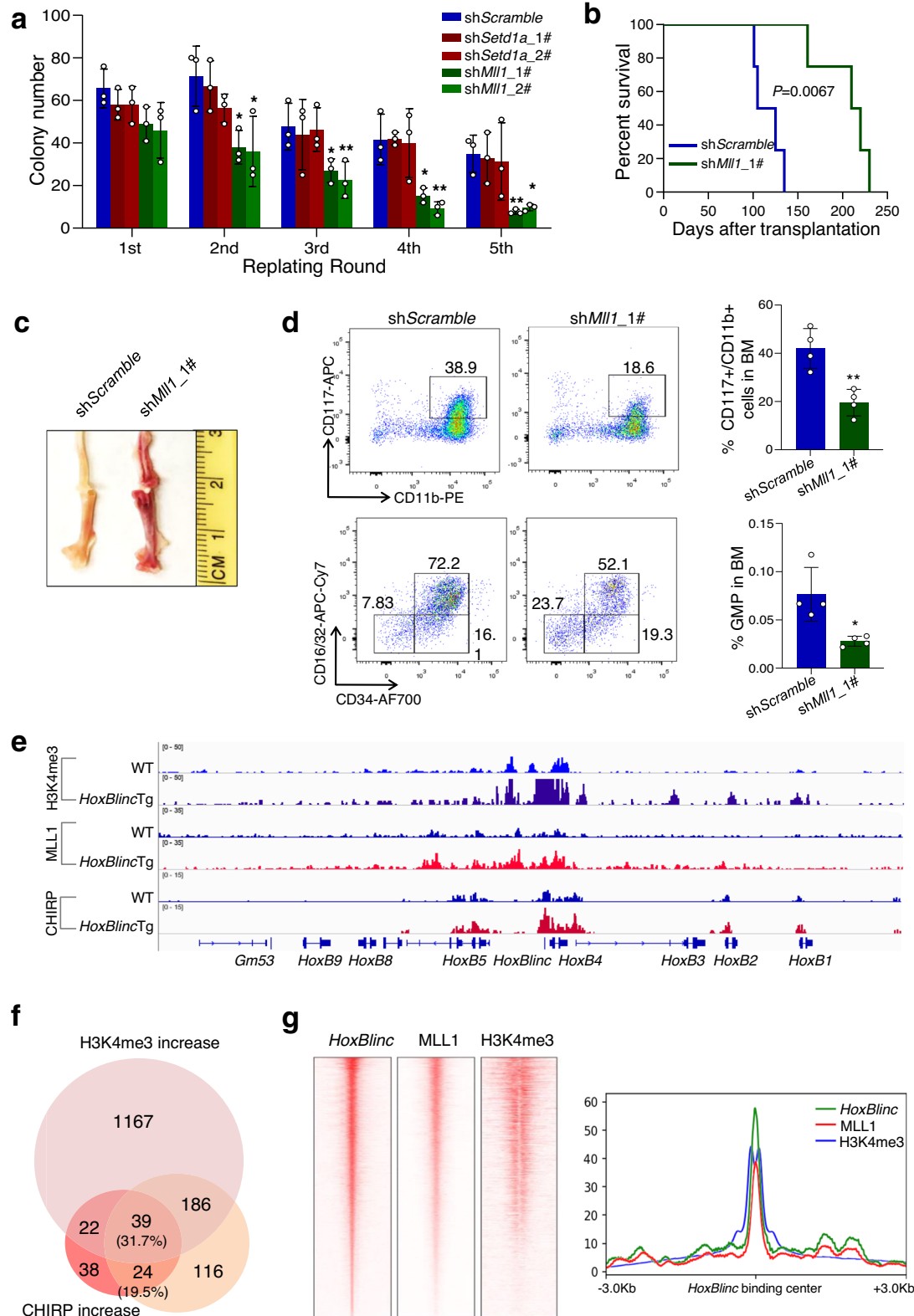

MLL1 recruitment and induces aberrant chromatin structure at NPM1c[+] signature genes.

LncRNAs are promising candidates for targeted cancer therapy, especially when they act as tissue-specific drivers of cancer. However, to the best of our knowledge, there are no therapeutic examples targeting lncRNAs in AML to date. *HOXBLINC* is highly expressed in HSCs. Given the profound impacts of *HOX-BLINC* overexpression on promoting HSC self-renewal and AML initiation, targeted therapeutics could be developed to suppress the overexpressed *HOXBLINC* lncRNA in *NPM1*-mutated AML. To

**Fig. 6 Recruitment of MLL1 is critical for _HoxBlinc_ overexpression mediated target gene expression and abnormal HSPC function. a** Number of colonies per 100 sorted GFP+ _HoxBlinc_Tg BM LSK cells transduced with Sh_Scramble_, sh_Setd1a_, or sh_Mll1_ lentivirus are shown (1st). Colonies were replated every 7 days for 4 times (2nd–5th). (Data from 3 independent experiments are presented as mean ± SD; *P < 0.05 and **P < 0.01 by two-tailed unpaired Student's t-test.) **b** Kaplan-Meier survival curve of recipient mice receiving Sh_Scramble_ or sh_Mll1_ LSK cells (Log-rank test was used for statistic analysis, n = 4 mice/group). **c** Gross appearance of femur dissected from mice transplanted with sh_Scramble_ or sh_Mll1_ LSK cells. **d** FACS analysis of BM CD11b+/CD117+ (_top_) and GMP (_bottom_) cell populations from recipient mice (4 mice/group) transplanted with sh_Scramble_ or sh_Mll1_ LSK cells. Data are presented as mean ± SD; *P < 0.05, **P < 0.01 by two-tailed unpaired Student's t-test. **e** ChIP-seq analysis of H3K4me3 (_top 2 tracks_) and MLL1 (_middle 2 tracks_), as well as ChIRP-seq analysis of _HoxBlinc_ (_bottom 2 tracks_) in the _HoxB_ loci of WT and _HoxBlinc_Tg Lin−c-Kit+ cells. **f** The overlap among genes with gain of _HoxBlinc_, H3K4me3, and MLL1 bindings to their promoter regions based on ChIRP-seq and ChIP-seq analyses. g Heatmap (_right_) and peaks density (_left_) to show the overlap of _HoxBlinc_ ChIRP-seq, MLL1, and H3K4me3 ChIP-seq.

support this, we show that knockdown of _HOXBLINC_, likely through the normalization of the aberrant NPM1c+ signature gene expressions, prolongs the survival of mice transplanted with _NPM1c+_ AML cells. In line with our findings, Qian et al. recently reported that CRISPR-Cas9-mediated specific DNA methylation at _DERARE_ attenuates _HOXB_ gene expression and alleviates leukemogenesis[28].

Our studies have provided convincing evidence that NPM1c+ upregulates a set of signature gene expression including _RUNX1_ gene via an increased _HOXBLINC_ expression. However, an important question remains, which is how _HOXBLINC_ is upregulated in NPM1c+ AML. Understanding of the underlying mechanisms by which _HOXBLINC_ itself is regulated in AML could provide insight into NPM1c+ -driven pathogenesis, but also lead to a strategy for the treatment of NPM1c+ AML via blocking _HOXBLINC_ activation. Although _RUNX1_ and _NPM1_ mutations are mutually exclusive in AML[29], RUNX1, a critical transcription factor in HSPCs and myelopoiesis, is frequently dysregulated in AML both genetically and transcriptionally. Instead of a unidirectional NPM1c+ -_HOXBLINC_-RUNX1 axis, NPM1c may also directly activate RUNX1 expression independent of _HOXBLINC_ leading to a positive feedback loop to control the _RUNX1_ and _HOXBLINC_ expression. Another scenario could be that cytosolic relocation of NPM1c+ alters CTCF-mediated AML genome topology-driven leukemic transcription networks. It has been reported that nuclear NPM1 associates with CTCF and acts as a nuclear matrix anchor for CTCF-mediated chromatin looping[30]. Indeed, CTCF boundaries were involved in defining transcriptionally active anterior HoxB domain including _HoxBlinc_ lncRNA in Hox gene activation during embryonic development[10]. Thus, it warrants further investigation of upstream signal/transcription networks leading to aberrant homeotic transcription program in AML leukemogenesis.

In summary, we show that _HOXBLINC_ overexpression is a critical event to drive leukemogenesis by establishing aberrant NPM1c+ signature gene expression program via controlling the MLL1 recruitment, chromatin domains, and promoter accessibility in _cis_ and _trans_ actions. Our studies, therefore not only provide molecular insights into the biology of HSC and _NPM1_-mutated AML, but also create a unique opportunity for the identification of drug targets for NPM1c+ AML.

## Methods

**Generation of the _HoxBlinc_ transgenic (Tg) mouse model**. All studies were conducted in accordance with the regulatory guidelines by the Institutional Animal Care and Use Committee (IACUC) at the UT Health San Antonio and University of Miami Miller School of Medicine. The full-length of mouse _HoxBlinc_ cDNA was cloned into HS321/45-vav vector[31] by SfiI and NotI so the transgene locates between the _Vav1_ promoter and _Vav1_ enhancer. This makes the transgene specifically express in hematopoietic system. The circular plasmid was then linearized by SacII into 2 fragments, only the longer fragment (Vav1 promoter-HoxBlinc-Vav1 enhancer, ~14 kb) was extracted and injected into the nuclear of zygotes of C57BL/6 mice while the shorter fragment (pBlueScript II SK backbone, ~3 kb) was discarded. Two _HoxBlinc_Tg founder mice were obtained by genotyping the tail genomic DNAs with P1 (forward primer, to detect both endogenous and transgenic _HoxBlinc_ gene) and P2 (reverse primer, to specifically recognize the transgenic _HoxBlinc_ gene), see

primer sequences in Supplementary Table 6. Transgenic mice showed a 589 bp band while the WT mice showed no band. Both founder mice were then crossed with WT C57BL/6 mice, the _HoxBlinc_ negative littermates were used as controls and the _HoxBlinc_Tg mice were used as experimental group throughout the study. These two lines of _HoxBlinc_Tg were analyzed separately. The m_HoxBlinc_-set1 and m_HoxBlinc_-set2 were used as real-time PCR primers to recognize both endogenous and exogenous _HoxBlinc_ lncRNA (Fig. 2d, Supplementary Table 6). The transgenic _HoxBlinc_ expression levels were also confirmed by RNA-seq analysis (Fig. 4a).

**Mouse housing conditions**. A 14-hour light/10-hour dark cycle is used. Researchers do not enter the mouse room during the dark cycle. Room temperatures were set to 21 °C with 40–60% humidity. Immunodeficient mice (NSG) were housed in the same room condition but in a separate room established for immunodeficient mice at the UT Health SA or University of Miami.

**Morphological and histological analyses of the hematopoietic organs**. PB was collected by tail vein bleeding and was subjected to an automated blood count (Hemavet System 950FS). PB smears were subjected to May-Grünwald-Giemsa staining for morphological and lineage differential analyses. Morphological evaluation of BM and spleen cells were performed on cytospins followed by May-Grünwald-Giemsa staining. For histopathological analyses, femurs were fixed in 10% Neutral Buffered Formalin and demineralized in a solution of 10% EDTA for 1–2 weeks, and the soft tissues such as spleen, lymph node, and liver were fixed in 10% Neutral Buffered Formalin. These fixed specimens were dehydrated with ethanol, cleared in xylenes, and then embedded in melted paraffin and allowed to harden. 5μm sections were cut and floated onto microscope slides. For routine assessment, slides were stained with hematoxylin and eosin (H&E staining). For MPO and hCD45 immunohistochemical staining, the tissue was rehydrated followed by heat-induced epitope retrieval, peroxidase, and serum blocking. Samples were then incubated with anti-MPO (R&D, #MAB3174, dilution: 1:500) or anti-hCD45 (Abcam, ab10559, dilution: 1:1000) antibody (overnight at 4 °C followed by staining with the biotinylated secondary antibody. Slides were visualized under a Nikon TE2000-S microscope. Images were taken by a QImaging camera and QCapture-Pro software (Fryer Company Inc.). Chemicals were obtained from Sigma (St. Louis, MO) unless otherwise indicated.

**Flow cytometry analysis, cell sorting, and colony assay**. The peripheral blood WBCs were obtained by treating peripheral blood with red blood cell lysis buffer (QIAGEN 1045722). BM, spleen, and peripheral blood WBC cells were stained with indicated flow antibodies (flow antibodies were listed in Supplementary Table 7, lineage flow antibody was diluted as 1:25, all other flow antibodies were diluted as 1:50). The flow data was collected by BD LSRII or LSR Fortessa and analyzed with FlowJo.V10 software. The LSK cells used for colony and replating assay were purified by flow sorting: total BM cells of 6–8 weeks old mice were pre-purified with lineage depletion beads (MiltenyiBiotec, Bergisch Gladbach, Germany) and then incubated with c-Kit, lineage, and Sca-1 antibodies (see Supplementary Table 7 for antibody information) and then sorted on BD FACS AriaII. IIThe purity of sorted LSK cells were routinely over 98%. The LSKs were incubated in methylcellulose medium (Methocult M3231) which was diluted with RPMI1640, supplemented with 30% FBS, 2% BSA, and a combination of cytokines (mG-CSF, 10 ng/mL; mIL-3, 5 ng/mL; mEPO, 4 U/mL; hTPO, 100 ng/mL; and mSCF, 100 ng/mL) for colony assay, colony numbers were counted at 7 days. For replating assay, the colonies were passaged sequentially every 7 days for 4 times. The gating strategy of Flow analysis and sorting were included in Supplementary Table 9.

**Competitive repopulation assay**. A total of $1 \times 10^6$ BM cells prepared by mixing $5 \times 10^5$ CD45.2 (WT or _HoxBlin_Tg) with $5 \times 10^5$ CD45.1 (B6.SJL) were injected into the tail veins of lethally irradiated (950 cGy) B6.SJL recipients (CD45.1). The contribution of CD45.1+ vs. CD45.2+ cells in the PB was monitored every month for 6 months after transplantation.

**Paired-daughter cell assay**. To examine the frequency of HSCs to undergo self-renewal and differentiation, we performed paired-daughter cell assays[32]. Single CD34−LSK cells from BM of WT and _HoxBlinc_Tg mice were clone-sorted into

96-well plates. The cells were maintained in RPMI1640 media supplemented with mSCF (100 ng/mL) and hTPO (50 ng/mL). After the first cell division, the two daughter cells were separated, one per well for an additional 12 days in the media supplemented with mSCF, hTPO, mEPO, mIL-3, and mG-CSF. The self-renewal and differentiation capabilities of cultured CD34−LSK cells were determined by morphological analyses of the progenies of the two daughter cells microscopically following May-Grünwald-Giemsa staining. A total of 192 single cells (two 96-well plates) were analyzed to calculate the percentage of symmetric/asymmetric cell divisions.

**Human AML samples and patient data analysis**. All human samples from healthy donors and patients with primary AML were obtained after informed consent following the guidelines of the Institutional Review Board of Pennsylvania State University College of Medicine (IRB protocol #2000-186 and #29252 EP) or the Institute of Hematology and Blood Disease Hospital, Tianjin, China. For *HOXBLINC* gene expression level analysis, BM low-density mononuclear cells (MNCs) were purified using Ficoll-Hypaque. RNA specimens extracted from MNCs of AML patients, MNCs of healthy controls or CD34+ cells of healthy controls were treated with RNase-free DNase to remove contaminating genomic DNA and first-strand cDNA was then synthesized. Real-time PCR was performed using Fast SYBR Green master mix. PCR amplifications were performed in triplicate with parallel measurements of human *GAPDH* (internal controls). The expression levels of *HOXBLINC* lncRNA were also assessed by analyzing a TCGA AML dataset from 179 AML patients (GEO accession number: GSE62944).

**RNA isolation, quantitative RT-PCR, as well as RNA-sequencing and data analysis**. Total RNAs were extracted and purified with the RNeasy mini-isolation kit according to the manufacturer's instructions (Qiagen, MD, USA). A total of 2 μg RNA was subjected to reverse-transcription with Superscript II Reverse Transcriptase (Invitrogen) and analyzed by a real-time PCR Detection System (Bio-Rad). Primer sequences for qPCR are listed in Supplementary Table 6 and key reagents are listed in Supplementary Table 7. For RNA-seq library generation, RNA sequencing library was prepared with the Illumina TruSeq mRNA sample preparation system kit (Cat# 20020594). In brief, total RNA samples were purified with purification beads, and then fragmented with the fragmentation buffer mix. Next, First strand cDNA was synthesized with the First Stand Synthesis Act D mix. Then, the second strand cDNA was synthesized with the Second Strand Marking Master Mix. After that, RNA-seq libraries were amplified and indexed with the adapter primers. Then, the quality of the library was tested with Qubit and Agilent Bioanalyzer. Final libraries were submitted to paired-end sequencing of 50 bp length on an Illumina HiSeq 3000. For RNA-seq data analysis, cutadapt (http://cutadapt.readthedocs.io, version 1.2.0) program was used to trim the adaptors and low quality reads from the RNA-seq raw data files[33]. Then, all of the filtered sequencing reads were processed and aligned to the mouse genome assembly (mm9) or human genome assembly (hg19) using TopHat (version 2.0) and Bowtie2[34–36]. Next, FPKM (paired-end fragments per kilobase of exon model per million mapped reads) was calculated for each gene and further normalized. The differential expression was determined according to the FPKM value and processed with the Cufflinks v2.2.1 and Cuffdiff[37]. Next, differentially expressed genes with greater than 2.0-fold were identified through comparing WT control and experimental groups according to the FPKM values. The scatter plot was generated according to the $\log_2$ transformation of the FPKM values, and the upregulated or downregulated genes with more than two folds changes were marked with blue dots, and no change genes were marked with red dots. Gene Ontology analysis was carried out with the Database for Annotation, Visualization and Integrated Discovery (DAVID) tool (https://david.ncifcrf.gov/, Version 6.8)[38]. Gene set enrichment analysis (GSEA) was conducted according to recommended parameters (http://software.broadinstitute.org/gsea/doc/GSEAUserGuideFrame.html) using gene sets obtained from the Molecular Signatures Database[39]. The normalized expression data tracks were loaded into the Integrated Genomic Viewer (IGV) for visualization. The sequence reads have been deposited in the NCBI GEO dataset (GSE115096). Key software and algorithms used are listed in Supplementary Table 7.

**RNA immunoprecipitation (RIP) assay**. The RNA-IP protocol was performed according to the previous reports[40,41]. In brief, OCI-AML3 and OCI-AML2 cells were harvested and washed with PBS, centrifuged and re-suspended in freshly RIP nuclear isolation buffer (1.28 M sucrose, 40 mM Tris-HCl pH 7.5, 20 mM MgCl₂, 4% Triton X-100), and then cell pellets were kept on ice for 20 min (with frequent mixing). Next, nuclei from cells were precipitated by centrifugation at 2,500 g for 15 min, and then re-suspended in freshly RIP lysis buffer (10 mM HEPES-KOH pH7, 150 mM KCl, 5 mM MgCl₂, 5 mM EDTA, 0.5% IGEPAL-CA-630, 0.5 mM dithreitol, 0.2 mg/mL Heparin, 100 U/mL RNse OUT, 100 U/mL Superase IN, protease inhibitor tablet adding before use). After that, chromatin shearing was performed with sonication. The suspension was centrifuged at 14,000 g at 4 °C for 10 min, and the supernatant was harvested. Next, the nuclear extracts were pre-cleared with rabbit IgG (Sigma) and Dynabeads™ Protein G (Thermo Scientific). Next, the nuclei extracts were incubated with 5 μg antibodies against LSD1 (Millipore, 07-705), Setd1a (Bethyl, A300-289A), and MLL1 (Novus Biologicals,

NB600-248) overnight at 4 °C. Immuno-complexes were captured by incubating with 50 μl Dynabeads™ Protein G (Thermo Scientific) for another 2 h at 4 °C. Next, the precipitant was washed with the ice-cold washing buffer (50 mM Tris-HCl pH 7.5, 150 mM NaCl, 1 mM MgCl₂, 0.05% IGEPAL-CA-630) supplemented with 0.02 mg/mL heparin. The RNA-protein complexes were eluted with 500 μL elution buffer (50 mM Tris pH 8.0, 100 mM NaCl, 10 mM EDTA, 1% SDS) for 10 min at 65 °C. The precipitated RNA was extracted and purified by TRIzol RNA extraction reagent, eluted with nuclease-free water, treated with DNaseI, and then endogenous RNA was subjected to the RT-qPCR analysis.

**Chromatin immunoprecipitation (ChIP)**. ChIP assay was performed as described previously[42]. Briefly, cells were cross-linked with 1% formaldehyde for 10 min at room temperature, and then quenched by addition of 125 mM glycine for 5 min at room temperature. The cell pellets were washed with ice-cold 1x PBS. Then, the cell pellets were re-suspended in ChIP lysis buffer (50 mM Tris-HCl, pH 8, 10 mM EDTA, 1% SDS). The lysates were sonicated with the Bioruprtor™ UCD200. After fragmentation, the lysates were centrifuged at 14,000 rpm for 10 min at 4 °C. The supernatant was diluted with ChIP buffer (20 mM Tris-HCl, pH 8.0, 2 mM EDTA, 1% Triton X-100, 150 mM NaCl). The 10% of remaining lysate was used as an input control. Sheared chromatin samples from $5 \times 10^6$ AML cells were immunoprecipitated with 5 μg anti-MLL1 (Novus Biologicals, cat# NB600-248) or 2.5 μg anti-H3K4me3 (Millipore, cat#04-745) overnight at 4 °C, separately. Then 50 μL Dynabeads™ Protein G (Thermo Fisher Scientific) was incubated with each ChIP sample for 2 h at 4 °C with rotation. The precipitated complexes were washed with salt (20 mM Tris, pH 8.0, 150 mM NaCl, 2 mM EDTA, 1% Triton X-100, 0.1% SDS), high salt (20 mM Tris, pH 8.0, 500 mM NaCl, 2 mM EDTA, 1% Triton X-100, 0.1% SDS) and lithium chloride (10 mM Tris, pH 8.0, 1 mM EDTA, 1% Triton X-100, 250 mM LiCl, 1% sodium deoxycholate) washing buffer, with an additional final wash in TE buffer (50 mM Tris-HCl, pH 8.0, 10 mM EDTA). Then, elution buffer (100 mM NaHCO₃, 1% SDS) was used to dissolve these precipitated complexes, and then were subjected to reverse crosslinking with 2 μL of 10 mg/mL proteinase K overnight at 65 °C. After reverse-crosslinking, the DNA samples were purified and then analyzed by RT-qPCR. The final results represent the percentage of input chromatin and error bars through triplicate experiments. The MLL1 and H3K4me3 ChIP-DNA libraries were prepared using Illumina's TruSeq ChIP Sample Preparation Kit according to the manufacturer's instructions (Cat# IP-202-1012). In brief, 10 ng ChIP DNA fragments were performed end repair using the End Repair Mix, and then purified with the AMPure XP beads. After that, 3′ ends of the ChIP DNA fragments were adenylated with the A-Tailing Mix, and then ligated with the adapter indices. After that, ChIP DNA fragments were amplified with the adapter primers. Then, the quality of these DNA libraries were tested with the Qubit and Agilent Bioanalyzer. Final libraries were submitted to paired-end sequencing of 50 bp length on an Illumina HiSeq 3000. The quality of the library was checked with Qubit and Agilent Bioanalyzer. Final libraries were submitted to paired-end sequencing of 100 bp length on an Illumina HiSeq 3000.

**Chromatin Isolation by RNA Immunoprecipitation (ChIRP) assay**. ChIRP assay was carried out to analyze the *HoxBlinc* lncRNA distribution in the genome according to our previously described with some modifications[43]. Briefly, 20 million cells were harvested and cross-linked in 20 ml of PBS buffer containing 1% glutaraldehyde (Sigma, Cat# G5882) at room temperature for 10 min, and then were quenched with a 1/10th volume of 1.25 M glycine at room temperature for 5 min. Next, these cross-linked cells were washed twice with chilled PBS, and per 100 mg of the pellet of cells were lysed in 1 ml lysis buffer (50 mM Tris-HCl pH 7.0, 10 mM EDTA, 1% SDS, PMSF, DTT, P.I. and SUPERase were added before use). Then, these lysates were sonicated 30 min using a Bioruptor (Diagenode) to prepare chromatin in a 4 °C water bath at the highest setting with 30 s ON, 30 s OFF pulse intervals. Next, the sheared chromatin was centrifuged and supernatant was diluted using hybridization buffer (750 mM NaCl, 1%SDS, 50 mM Tris-HCl 7.0, 1.0 mM EDTA, 15% Formamide, add DTT, PMSF, P.I, and SUPERase-in fresh). After that, the diluted supernatant was hybridized with 100 pmol of biotinylated DNA probes targeting *HoxBlinc* or *LacZ* (sequences are listed in Supplementary Table 6) and incubated for 4 h at 37 °C with shaking, and then added 100 μl of Streptavidin-magnetic C1 beads (Invitrogen) for each sample for 30 min at 37 °C. Then, these precipitant were washed 5 times with washing buffer (2x SSC, 0.5% SDS). RNA fraction was extracted from ChIRP samples with TRIzol reagent and then subjected to analyze the *HoxBlinc* retrieval by RT-qPCR, and *β-actin* gene was used as a negative control. DNA fraction was extracted from ChIRP samples with Phenol:Chloroform:Isoamyl Alcohol (25:24:1, v/v), and then precipitated with 3 μL of 20 mg/mL glycogen, 1/10th volume of 3 M sodium acetate (pH 5.2), and 2.5 volumes of 100% ethanol. Then ChIRP-DNA library was prepared for ChIRP-seq. Libraries were constructed using Illumina's TruSeq ChIP Sample Preparation Kit according to the manufacturer's instructions (Cat# IP-202-1012). In brief, 10 ng ChIRP DNA fragments were performed end repair using the End Repair Mix, and then purified with AMPure XP beads. After that, 3' ends of the ChIRP DNA fragments were adenylated with A-Tailing Mix, and then ligated with the adapter indices. After that, ChIRP DNA fragments were amplified with the adapter primers. The quality of these ChIRP DNA libraries were examined with Qubit and Agilent Bioanalyzer. Final ChIRP libraries were submitted to the paired-end sequencing of 100 bp length on an Illumina HiSeq 2500.

**ChIP-seq and ChIRP-seq data analysis.** The cutadapt (http://cutadapt.readthedocs.io, version 1.2.0) program was used to trim the adaptors and low quality reads from the ChIP-seq or ChIRP-seq raw data[33]. These Cutadapt-filtered reads were aligned to mouse reference genome (mm9) using Bowtie2 with default parameters[35], and the quality of these trimmed data was evaluated by the FastQC program[44]. After alignment, samtools program was employed to convert the SAM files into BAM files and sorted these BAM files[45]. Next, peak calling was performed using peak calling algorithm MACS2[46]. Peaks were transformed to the visualized files (bigwig format) with deepTools[47], including control and experimental data-sets. All sequencing tracks were visualized using the Integrated Genomic Viewer software (IGV)[48]. Peaks were annotated with"annotatePeaks.pl" program with HOMER package[49]. Differential Peaks calling was performed with getDifferentialPeaks program in HOMER software. For ChIRP-seq binding motif analysis, the de novo motif analysis was performed by the "findmotifsgenome.pl" from HOMER motif discovery algorithm[49]. The *HoxBlinc* bound regions associated genes and pathways were analyzed and annotated by the Gene Ontology (GO) analysis with the Database for Annotation, Visualization and Integrated Discovery (DAVID) tool (https://david.ncifcrf.gov/, Version 6.8)**[37]. Each GO term with a p-value more than $1 \times 10^{-3}$ is used for cutoff (threshold: $10^{-3}$). All genomics datasets were deposited in the NCBI GEO under accession number (GSE115096).

**Circular Chromosome conformation capture (4C) assays.** The 4C-seq assay was performed as previously described[50] with minor modifications. In brief, $2 \times 10^6$ cells were cross-linked with 1% formaldehyde for 10 min and the reaction was quenched with 0.125 M glycine for 5 min at room temperature. Cells were washed twice with cold PBS, and re-suspended in the digestion buffer containing 0.3% SDS overnight at 37 °C with shaking. After that, 2% Triton X-100 was added to sequester SDS and incubated for 1.5 h at 37 °C with shaking. Then, the chromatin was digested with 400U of BglII enzyme (NEB) at 37 °C overnight. 1.6% SDS buffer was added to stop the digestion at 65 °C for 20 min. After that, the digested chromatin was diluted in T4 DNA ligation buffer (NEB) containing 1% Triton X-100 and incubated at 37 °C for 1.5 h with shaking. Next, 800U of T4 DNA ligase (NEB) was used to ligate these digested chromatin at 16 °C for 3 days followed by 1 hr at room temperature. Then, reverse crosslinking was carried out by adding 200 μg of Proteinase K (Invitrogen) and incubated at 65 °C overnight. Next, the 3 C DNA was subjected to phenol:chloroform and extracted with 100% ethanol, and then dissolved with ddH$_2$O. For the second digestion, the first ligates was digested with 300U of NlaIII (NEB) at 37 °C overnight with shaking. The reaction was stopped by adding SDS to a final concentration of 1.6% at 65 °C for 20 min. The digested chromatin/DNA was ligated in ligation buffer containing 6000 U of T4 DNA ligase at 16 °C overnight. After ligation, the 4 C DNA was extracted by phenol-chloroform, and purified by Qiagen PCR kit and amplified by inverse PCR using bait-specific primers. The invert PCR products were cloned into pGEM®-T Easy Vector Systems (Promega) for Sanger sequencing. 4C-seq Libraries were constructed by adding barcoded Illumina adapters to the 5′ end of each primer (Supplementary Table 6). PCR reactions were performed using the Expand Long Template PCR System (Roche), and DNA libraries were purified and quantified before sequencing. The bar-coded DNA libraries were sequenced as 150 bp pair-end reads using the Illumina Nextseq500 platform. For data analysis, cutadapt (http://cutadapt.readthedocs.io, version 1.2.0) program was carried out to trim the 4C-seq raw data and remove the adaptors and low quality reads[33]. Filtered reads were aligned to the reference mouse genome (build mm9) with Bowtie2 2.2.9[51]. 4C-seq data was analyzed using the 4cseq_pipeline[52] and normalized using DESeq2[53]. Statistical analysis for differential interactions between genotypes was performed using DESeq2. Spearman correlation of each genotype was performed using R[54]. The 4C-sequencing sequence reads have been deposited in the NCBI GEO database (GSE115096).

**Transposase-Accessible Chromatin using sequencing (ATAC-seq) assay.** ATAC-seq assay was used to analyze the genome chromatin accessibility in different experimental conditions as described previously[12,55]. In Brief, $5 \times 10^4$ cells were collected for library preparation. Cells were washed with pre-cold phosphate buffered saline (PBS) and re-suspended in lysis buffer containing 10 mM Tris-HCl (pH 7.4), 10 mM NaCl, 3 mM MgCl$_2$, 0.1% NP-40. After washing with cold 1x PBS buffer, the cell pellets were re-suspended in fragment buffer and then fragmented with Tn5 Transposes for transposition reaction at 37 °C for 30 min. Then these DNA fragments were extracted and purified using the MinElute Kit (QIAGEN). The preparation of library fragments were amplified using 1x NEB next PCR master mix and 1.25 μM indexed Nextra PCR primers (Supplementary Table 6) with following PCR conditions: 72 °C for 5 min, 98 °C for 30 s, followed by thermocycling at 98 °C for 10 s, 63 °C for 30 s and 72 °C for 1 min. After amplification, the eluted DNA was used in a quantitative PCR (qPCR) reaction to estimate the optimum number of amplification cycles. Libraries were quantified using qPCR (Kapa Library Quantification Kit for Illumina, Roche), and AMPure XP beads (Beckman Coulter) was employed to purify the libraries, and then the quality of the DNA library was examined by Agilent Bioanalyzer 2100 prior to sequencing with $2 \times 100$ bp paired-end reads on an Illumina HiSeq 2500. Each sample includes two replicates for statistical analysis.

**ATAC-seq analysis.** ATAC-seq assay was carried out and analyzed with two biological replicates according to our previous reports[12,55]. Briefly, all of the raw

data files were filtered through cutadapt (http://cutadapt.readthedocs.io, version 1.2.0) to remove adaptors and low quality reads[33]. These filtered reads were aligned to the mouse genome (mm9) using Bowtie2 with default parameters (version Bowtie 2/2.2.6)[35], and the quality of these trimmed data was evaluated by FastQC program (version 0.11.8)[44]. PCR duplicates were removed using Picard MarkDuplicates (version 2.0.1), and mitochondrial reads were removed with samtools[56]. ENCODE blacklist regions were filtered (https://sites.google.com/site/anshulkundaje/projects/blacklists). For quality control, 50 million reads with paired-end sequencing was used for each sample. In addition, the alignment rate of each replicate is more than 95% through removing the unaligned reads. Third, the mitochondrial-related reads and PCR duplicates reads were removed from total reads after alignment. Finally, non-uniquely aligned reads were filtered based on MAPQ scores with samtools (MAPQ > 30), and plotPCA from BiocGenerics package in R package (R/3.6.1) was carried out to identify the variance between control and experimental groups. Moreover, fragSizeDist from ATACseqQC package in R package was used to show the fragment size distribution in control and experimental groups. After alignment and trimming, samtools (version 1.8.0) were used to convert the SAM files into BAM files and sorted the BAM files for further analysis[45]. Peak calling was performed using peak calling algorithm MACS2 with parameters ("-g mm -p 1e-9 –nolambda -f BAMPE –nomodel –shiftsize=100 --extsize 200")[46]. The visualizable bigWig files of fragment or read coverages were generated with bedGraphToBigWig program, including control and experimental datasets (https://www.encodeproject.org/software/bedgraphtobigwig/). All sequencing tracks were viewed using the Integrated Genomic Viewer (IGV/2.4.19)[48]. Peaks were annotated with the command "annotatePeaks.pl" from HOMER package (version 4.10)[49] and GREAT[57]. Next, the differential binding sites between two peak files were calculated with DEseq2 (Benjamini-Hochberg adjusted $p < 0.05$; FoldChange≥2), including control and experimental groups with C + G normalized and "reads in peaks" normalized data[58]. The de novo motif analysis was performed by the "findmotifsgenome.pl" from the HOMER package[49]. For each genomic feature (peaks or chromVAR annotation), the chromatin accessibility median deviation z-score (for chromVAR features) or fragment counts (for peaks) were examined in control and experimental groups with chromVAR package in R language[59,60]. Overall similarity between the replicates of ATAC-seq global chromatin accessibility signatures was carried out with Pearson's correlation coefficient and Pearson's $\chi^2$-test. All genomics datasets were deposited in the NCBI GEO under accession number (GSE115096).

**dCas9-mediated inactivation of *HoxBlinc* in AML cells.** To generate the CRISPRi sgRNA vector, sgRNA targeting the promoters of *HOXBLINC* were designed using the Zhang laboratory web tool (http://crispr.mit.edu), and sgRNA was subcloned into the pLKO5.sgRNA.EFS.GFP vector (Addgene#57822). OCI-AML3 AML cells were co-transfected with the gRNA plasmids encoding GFP and the repressive plasmid encoding dCas9-KRAB (pHR-SFFV-dCas9-BFP-KRAB, Addgene plasmid #46911), and then these cells were cultured for another 48 h. After that, OCI-AML3 AML cells were subjected to 2 μg/mL of puromycin for another 48 h, and then live cells were sorted with GFP by FACS. Finally, RNAs from these positive cells were extracted with TRIzol™ reagent, and then the gene expression level was determined by RT-qPCR.

**Doxycycline (DOX) inducible sh*HOXBLINC* in AML cells.** Human shRNA targeting *HOXBLINC* was designed using the Thermo Fisher web tool (https://rnaidesigner.thermofisher.com/rnaiexpress/), and shRNA was cloned into the doxycycline (DOX) inducible pTRIPz vector (dharmacon.horizondiscovery.com). Then OCI-AML3 cells were infected with DOX inducible pTRIPz vector and then cultured for 48 h. After that, cells were selected with 2 μg/ml of puromycin for another 48 h, and the live cells were cultured in the standard culture medium of OCI-AML3 cell line. After that, cells were treated with the DOX at a concentration of 2 μg/mL for 3 days. In addition, refresh or add the fresh culture medium to maintain the doxycycline concentration every 48 h. Then, RNAs from these doxycycline inducible cells were extracted and purified with the TRIzol™ reagent, and then proceeded to evaluate the knockdown efficiency by RT-qPCR.

**Lentiviral transduction.** The RNAi Consortium (TRC)-based short hairpin RNA lentiviral vector (Sigma, Horizon Discovery, or lab-owned) were transfected into HEK 293 T cells (ATCC) together with pMD2.G (Addgene) and psPAX2 (Addgene) to package viruses. The virus supernatant was collected from 16 h to 48 h after transfection and then purified with Lenti-X™ Concentrator (Takara). The Lin⁻ cells from mice BMs were transduced with the packaged viruses for 24 h in plain IMDM added with mSCF (100 ng/mL) and then sorted for GFP⁺ LSK cells. Sorted LSK cells were collected into different dishes for colony-forming assays or transplantation.

**Xenotransplantation of OCI-AML3 AML cells or AML patient BM cells.** Adult NOD.Cg-Prkdc^scid Il2rg^tm1Wjl/SzJ (NSG) mice (6–8 weeks old) were pretreated with 280 cGy total body irradiation, then $5 \times 10^5$ viable OCI-AML3 cells (in 300 μl PBS) or $1 \times 10^6$ patient BM cells (in 300 μl PBS) were injected into tail veins of the recipient NSG mice for transplantation. After transplantation, the recipients mice were administered with (for OCI-AML3 cells stably expressing inducible *shHOXBLINC* to induce *HoxBlinc* KD) or without (for AML cells with dCas9-mediated *HoxBlinc* inactivation) doxycycline in the drinking water (Sigma D-9891, 1 mg/ml,

1% sucrose, newly prepared every other day) until being sacrificed, and daily monitored for symptoms (ruffled coat, hunched back, weakness, and reduced motility) and survival time. For each set of xenotransplantation, recipient mice of all groups were killed and analyzed on the same day when any group of recipients exhibited a moribund condition. Human CD45 chimerism in the BM, spleen cells, and PB WBC were analyzed by flow cytometry, histological assay, IHC for hCD45 were also performed as described above.

**Quantification and statistical analysis.** Differences between experimental groups were determined by the Student's t-test or analysis of variance (ANOVA) followed by Newman-Keuls multiple comparison tests as appropriate. $P < 0.05$ is considered significant. For in vivo experiments, the sample size chosen was based on the generalized linear model with Bonferroni multiple comparison adjustments; with the proposed sample size of at least five mice/group/genotype. Animals were randomly assigned to each study. For all in vitro experiments, at least three independent experiments with more than three biological replicates for each condition/genotype were performed to ensure adequate statistical power.

**Reporting summary.** Further information on research design is available in the Nature Research Reporting Summary linked to this article.

## Data availability

Sequencing data of RNA-seq, ATAC-seq, ChIP-seq and 4C-seq in Figs. 1, 4, 5, 6 & Supplementary Figs. 1, 2, 6, 7, 9 were deposited in the Gene Expression Omnibus (accession number GSE115096). RNA-seq data of AML patient samples were retrieved from The Cancer Genome Atlas (GEO accession number GSE62944). NCBI GEO public database (GSM918748) were used for CTCF binding site analysis of Fig. 5 and Supplementary Fig. 7. Source Data file is provided with this paper. All biological material were either directly commercially available or are available upon request from the lab (see Supplementary Table 7). Supplemental Information Supplemental information includes 9 figures and 9 tables. All other data are available from the authors. Source data are provided with this paper.

## Code availability

There is no custom R code in this study. The source of software and algorithms are included in Supplementary Table 7.

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

## Acknowledgements

This work was supported by grants from the National Institute of Health (R01HL141950 to M.X. and S.H., R01CA172408 to M.X. and F-C.Y., NIH R00CA178191 and R01DK121831 to O.A.G., and R01DK110108 to S.H.), the Cancer Prevention & Research Institute of Texas (CPRIT RP200242 to M.X.), as well as the Florida Academic Cancer Center Alliance (FACCA) Award (S.H. and M.X.). We also thank Rachael Mills at the Penn State Hershey College of Medicine for editing the manuscript.

## Author contributions

G.Z., H.L., Y.Q. F-C.Y., S.H., and M.X. conceived and designed experiments. G.Z., H.L, S.C., Q.L., and Y.G. performed experiments. H.L. performed bioinformatics and statistical analysis for mouse NGS data. J.X., Z.Z., R.F., A.S., D.C., B.X., M.R., and W.L. acquired and analyzed primary human patient samples and performed TCGA AML data analysis. H.N analyzed the histological slides from mouse tissues. Y.F. and O.A.G. provided *Npm1c*+ knock in mouse and related samples. M.X. and S.H. wrote the original draft, O.A.G., S.D.N., and F-C.Y. revised and edited the manuscript.

## Competing interests

The authors declare no competing interests.

## Additional information

[1]Department of Molecular Medicine, University of Texas Health San Antonio, San Antonio, TX 78229, USA. [2]Department of Biochemistry and Molecular Biology, University of Miami Miller School of Medicine, Miami, FL 33136, USA. [3]Department of Pediatrics, Pennsylvania State University College of Medicine, Hershey, PA 17033, USA. [4]Department of Pharmacology and Therapeutics, University of Florida College of Medicine, Gainesville, FL 32610, USA. [5]Department of Molecular and Cellular Biology, Dan L. Duncan Cancer Center, Baylor College of Medicine, Houston, TX 77030, USA. [6]Division of Hematology/Oncology, Department of Medicine, Pennsylvania State University College of Medicine, Hershey, PA 17033, USA. [7]Department of Hematology, The First Affiliated Hospital of Xiamen University, Xiamen 361026 Fujian, China. [8]Department of Hematology and Oncology, Tianjin Medical University Cancer Institute and Hospital, National Clinical Research Center for Cancer, Key Laboratory of Cancer Prevention and Therapy, Tianjin 300060, China. [9]Department of Hematology, Nanfang Hospital, Southern Medical University, Guangzhou 510515 Guangdong, China. [10]Department of Pathology, Wayne State University School of Medicine, Detroit, MI 48201, USA. [11]Penn State Cancer Institute, Pennsylvania State University College of Medicine, Hershey, PA 17033, USA. [12]Department of Cell System & Anatomy, University of Texas Health San Antonio, San Antonio, TX 78229, USA. [13]Department of Medicine, University of Texas Health San Antonio, San Antonio, TX 78229, USA. [14]Mays Cancer Center, University of Texas Health San Antonio, San Antonio, TX 78229, USA. [15]Department of Cellular & Molecular Physiology, Pennsylvania State University College of Medicine, Hershey, PA 17033, USA. [16]Sylvester Comprehensive Cancer Center, University of Miami Miller School of Medicine, Miami, FL 33136, USA. [17]Division of Computational Biomedicine, Department of Biological Chemistry, School of Medicine, University of California, Irvine, CA 92697, USA. [18]Department of Medicine, University of Miami Miller School of Medicine, Miami, FL 33136, USA. [19]These authors contributed equally: Ganqian Zhu, Huacheng Luo. [20]These authors jointly supervised: Suming Huang, Mingjiang Xu. ✉email: shuang4@pennstatehealth.psu.edu; xum1@uthscsa.edu

