## [Peer Review File · Nature Communications]

REVIEWER COMMENTS

Reviewer #2 (Remarks to the Author):

Fig. 1I is a new type of experiments that need more details. What were the blast % of these three donor AMLs? The authors should show evidence that the engraftment was from human AML cells but not multi-lineage human hematopoietic cells, and clarify whether the deaths of animals were caused by leukemia. In addition, the information of the three patients in Fig. 1I was not included in Table S1.

Reviewer #4 (Remarks to the Author):

Authors reported their studies of HOXBLINEC in NPM1c related AML development. The presented work is very well done. Data are in high quality. Writing is clear. Since NPM1 is the most commonly mutated gene in AML, this report is highly significant and with broad interest to understand molecular mechanism of how NPM1c promotes leukemogenesis. Furthermore, they have addressed most comments of previous reviewers. Their work provides convincing data to support their conclusion that NPM1c expression increases HOXBLINEC and HOXBLINEC upregulated NPM1c signature gene expression. They also addressed whether HOXBLINEC was important for MLL related leukemia. An important question is how HOXBLINEC is upregulated in AML with NPM1 mutation. Based on their point to point responses, authors have partially investigated whether HOX family transcription factors regulate HOXBLINEC expression. Since RUNX1 and NPM1 mutations are mutually exclusive in AML, RUNX1 may enhance HOXBLINEC expression. Instead of a unidirectional regulation, NPM1c may also activate RUNX1 expression independent of HOXBLINEC and results in a positive feedback loop to control the RUNX1-HOXBLINEC expression. Therefore, authors may include additional discussion along this line.

Reviewer #2:

1. What were the blast % of these three donor AMLs? In addition, the information of the three patients in Fig. 1I was not included in Table S1.

Response: We thank the reviewer for pointing this important information out. The characteristics of the three AML patients have now been added into Supplemental Table S1.

2. Fig. 1I is a new type of experiments that need more details. The authors should show evidence that the engraftment was from human AML cells but not multi-lineage human hematopoietic cells, and clarify whether the deaths of animals were caused by leukemia.

Response: We appreciate the reviewer's insightful comments. As suggested by Reviewer #2, we performed an additional transplantation experiment using the #1315 AML cells (*NPM1c⁺;FLT3wt*) with (*HOXBLINEC_i*) or without (control) *CRISPR-dCas9-KRAB* mediated silencing of *HOXBLINEC* expression. We chose the #1315 AML patient, simply because we still have sufficient frozen BM cells. Based on the number of available cells, we were able to transplant 5×10^5 control or *HOXBLINEC_i* #1315 AML cells into two NSG recipients each. Consistent with the #1315 AML cell xenograft results shown in the previous version of manuscript, mice receiving control #1315 AML cells became moribund 18 days after transplantation. Therefore, both groups of recipient mice receiving either control or *HOXBLINEC_i* #1315 AML cells were sacrificed 18 days after transplantation for examination of human AML cell engraftment and AML development by combined FACS, immunohistochemical and morphological (cytospin) analyses on the BM cells. These analyses confirmed that *HOXBLINEC_i* dramatically decreased the hCD45⁺ cell chimeras in the BM as compared to control AML cells in this *NPM1c⁺* #1315 AML cell xenografts (Figures 1J & S2F). Using a panel of antibodies against hCD34, hCD33, hCD19 and hCD3, the engrafted human cells in both control and *HOXBLINEC_i* recipients were positive for hCD33 and negative for CD19 and CD3, with a small fraction (~4%) being CD34⁺CD33^{low} and over half being CD34^{low/+}CD33⁺ (Figure S2F). Immunohistochemical analyses of BM sections using both H&E staining and anti-human CD45 immunostaining (*brown*) revealed plentiful human CD45⁺ cells (Figure 1J), and BM cytospin analyses showed significant proportions of myeloid blasts (Figure S2G). These data demonstrated that the engraftment was from human AML cells but not multi-lineage human hematopoietic cells, and also clarified that the moribund state/death of animals were caused by AML development and progression. These data are included in Figures 1 and S2F,G (see Page 6, lines 17-29).

Reviewer #4:

An important question is how HOXBLINEC is upregulated in AML with NPM1 mutation. Based on their point to point responses, authors have partially investigated whether HOX family transcription factors regulate HOXBLINEC expression. Since RUNX1 and NPM1 mutations are mutually exclusive in AML, RUNX1 may enhance HOXBLINEC expression. Instead of a unidirectional regulation, NPM1c may also activate RUNX1 expression independent of HOXBLINEC and results in a positive feedback loop to control the RUNX1-HOXBLINEC expression. Therefore, authors may include additional discussion along this line.

Response: We appreciate the reviewer's very thoughtful comments. We, accordingly, revised the discussion section to reflect this important point as follows.

Our studies have provided convincing evidence that NPM1c+ upregulates a set of signature gene expression including *RUNX1* gene via an increased *HOXBLOC* expression. However, an important question remains, which is how *HOXBLOC* is upregulated in NPM1c+ AML. Understanding of the underlying mechanisms by which *HOXBLOC* itself is regulated in AML could provide insight into NPM1c+-driven pathogenesis, but also lead to a novel strategy for the treatment of NPM1c+ AML via blocking *HOXBLOC* activation. Although *RUNX1* and *NPM1* mutations are mutually exclusive in AML (ref #29) *RUNX1*, a critical transcription factor in HSPCs and myelopoiesis, is frequently dysregulated in AML both genetically and transcriptionally. Instead of a unidirectional NPM1c+-*HOXBLOC*-*RUNX1* axis, NPM1c may also directly activate *RUNX1* expression independent of *HOXBLOC* leading to a positive feedback loop to control the *RUNX1* and *HOXBLOC* expression. Another scenario could be that cytosolic relocation of NPM1c+ alters CTCF-mediated AML genome topology-driven leukemic transcription networks. It has been reported that nuclear NPM1 associates with CTCF and acts as nuclear matrix anchor for CTCF-mediated chromatin looping (ref #30). Indeed, CTCF boundaries were involved in defining transcriptionally active anterior HoxB domain including *HoxBlinc* lncRNA in Hox gene activation during embryonic development (ref #10). Thus, it warrants further investigation of upstream signal/transcription networks leading to aberrant homeotic transcription program in AML leukemogenesis. The discussion section has been revised accordingly (see Page 16, the last paragraph).

REVIEWERS' COMMENTS

Reviewer #2 (Remarks to the Author):

The authors have addressed the reviewer's comments.